# Hungry Hungry Hippos: Towards Language Modeling with State Space Models

**Daniel Y. Fu**[\*†]**, Tri Dao**[\*†]**, Khaled K. Saab**[†]**, Armin W. Thomas**[†]**, Atri Rudra**[‡]**, Christopher Ré**[†]
† Stanford University, ‡ University at Buffalo, SUNY
`{danfu,tridao}@cs.stanford.edu,{ksaab,athms}@stanford.edu,`
`atri@buffalo.edu,chrismre@cs.stanford.edu`

## Abstract

State space models (SSMs) have demonstrated state-of-the-art sequence modeling performance in some modalities, but underperform attention in language modeling. Moreover, despite scaling nearly linearly in sequence length instead of quadratically, SSMs are still slower than Transformers due to poor hardware utilization. In this paper, we make progress on understanding the expressivity gap between SSMs and attention in language modeling, and on reducing the hardware barrier between SSMs and attention. First, we use synthetic language modeling tasks to understand the gap between SSMs and attention. We find that existing SSMs struggle with two capabilities: recalling earlier tokens in the sequence and comparing tokens across the sequence. To understand the impact on language modeling, we propose a new SSM layer, H3, that is explicitly designed for these abilities. H3 matches attention on the synthetic languages and comes within 0.4 PPL of Transformers on OpenWebText. Furthermore, a hybrid 125M-parameter H3-attention model that retains two attention layers surprisingly outperforms Transformers on OpenWebText by 1.0 PPL. Next, to improve the efficiency of training SSMs on modern hardware, we propose FLASHCONV. FLASHCONV uses a fused block FFT algorithm to improve efficiency on sequences up to 8K, and introduces a novel state passing algorithm that exploits the recurrent properties of SSMs to scale to longer sequences. FLASHCONV yields 2× speedup on the long-range arena benchmark and allows hybrid language models to generate text 2.4× faster than Transformers. Using FLASHCONV, we scale hybrid H3-attention language models up to 2.7B parameters on the Pile and find promising initial results, achieving lower perplexity than Transformers and outperforming Transformers in zero- and few-shot learning on a majority of tasks in the SuperGLUE benchmark.

## 1 Introduction

State space models (SSMs) have achieved state-of-the-art sequence modeling performance in domains ranging from time series analysis (Gu et al., 2022a) to audio generation (Goel et al., 2022). However, they have yet to match the performance of Transformers on language modeling, often underperforming Transformers by multiple points in perplexity (Gu et al., 2022a). An natural question is whether this gap in performance is due to inherent inductive biases and capabilities in attention (Edelman et al., 2022; Olsson et al., 2022), or whether it is a function of the significant organizational resources that have been spent training and tuning large attention-based language models (Chowdhery et al., 2022; Hoffmann et al., 2022; Zhang et al., 2022), as well as specialized hardware support for attention, ranging from tensor cores (NVIDIA, 2017) to transformer chips (NVIDIA, 2022b; Kao et al., 2021).

We take first steps towards answering these questions in this paper. First, we use synthetic language modeling tasks to show that there is an expressivity gap between SSMs and attention. Using our insights, we design a new SSM layer that nearly matches attention in language modeling. Second, we propose better hardware-aware algorithms for SSMs that allow them to take advantage of modern accelerators—and run faster than attention.

**Understanding the Expressivity Gap.** To understand the gap between SSMs and attention, we draw on synthetic language modeling tasks that have been proposed as a mechanistic basis for in-context learning in Transformers (Olsson et al., 2022) These synthetic languages focus on the ability to manipulate text—recalling tokens from earlier time steps, or comparing tokens from different points in a sequence. We find that existing SSMs struggle to model these synthetic languages. To probe how important these

---

*Equal Contribution. Order determined by coin flip.

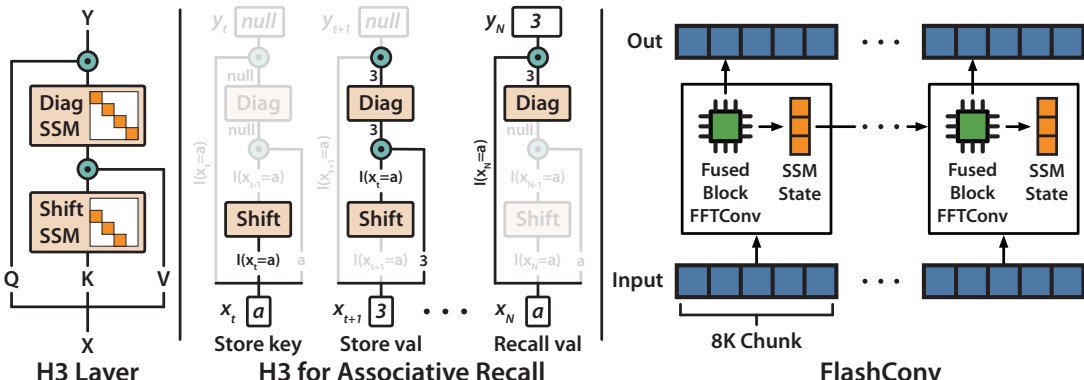

Figure 1: Left: H3 stacks two discrete SSMs with shift and diagonal matrices and uses multiplicative interactions between input projections and their outputs to model comparisons between points in a sequence. Middle: H3 can perform associative recall—which is easy for attention, but not existing SSMs. Right: FLASHCONV uses a new state-passing algorithm over fused block FFTConv to increase hardware efficiency of SSMs, allowing H3 to scale to billion-parameter models.

skills are for language modeling, we propose H3 (Hungry Hungry Hippo), a new SSM-based layer designed to solve these language modeling tasks. H3 stacks two SSMs, with multiplicative interactions between their outputs and input projections. The SSMs allow H3 to keep a log of tokens (to recall them later), while the multiplicative interactions allow for comparisons across the sequence.

H3 matches attention on the synthetic languages and almost closes the gap with Transformers on language modeling—coming within 0.4 perplexity of Transformers on OpenWebText (compared to 3.4 ppl for existing SSMs—even those explicitly designed for language modeling (Mehta et al., 2022)). Furthermore, a simple hybrid H3-attention model that retains two attention layers surprisingly *outperforms* Transformers on OpenWebText by 1.0 perplexity. To further evaluate H3 on language modeling, we train 125M-, 355M-, 1.3B-, and 2.7B-parameter hybrid H3-attention language models on the Pile (Gao et al., 2020), using hyperparameters from GPT-3 (Brown et al., 2020). These hybrid models outperform Transformer-based language models of the same size in perplexity, and match or outperform them on a majority of tasks in the SuperGLUE benchmark in zero- and few-shot learning. Since the SSM layers in these hybrid models admit a recurrent view, they can also perform $2.4\times$ faster inference than Transformers.

**Scaling SSMs.** Next, we improve the efficiency of SSMs on modern hardware, to reduce the hardware barrier between attention and SSMs. SSMs scale nearly linearly in sequence length instead of quadratically like attention, but still run slower on modern hardware due to poor hardware utilization. To close this gap, we propose FLASHCONV, a hierarchical algorithm for computing SSMs, inspired by IO-Aware attention (Dao et al., 2022b). The technical challenge is that SSMs require a FFT-based convolution over the input sequence, which requires an FFT, pointwise multiply, and inverse FFT. When implemented in cuFFT (NVIDIA, 2022a), this operation incurs expensive GPU memory reads/writes, and cannot utilize the specialized matrix multiply units available on modern hardware[1]. To use specialized matrix multiply units, we appeal to classical techniques that split the FFT into blocks and compute it using a series of matrix multiplications. Combined with kernel fusion, this "block" FFT solution increases hardware efficiency, but only as long as the sequence length can fit into GPU SRAM (on-chip memory, analogous to L1 cache on the CPU)—up to sequence length 8K on modern A100.

To scale to sequences longer than 8K, we propose a *state passing* algorithm (Figure 1 right), specialized to SSMs. The key insight is that we can use the recurrent properties of SSMs to process the input in chunks—as long as we keep track of an additional state vector. The state passing algorithm splits the input into the largest chunks that can fit into GPU SRAM, efficiently computes the FFT-based convolution using block FFT, and updates an intermediate state to start the next chunk. Using this state-passing algorithm, FLASHCONV can scale SSMs to *any* sequence length—even longer than can fit on GPU SRAM at once—while maintaining a *near linear* compute complexity. FLASHCONV sets state-of-the-art speed on long range arena using S4 (Gu et al., 2022a), outperforming Transformers by $5.8\times$ and previous S4 models by $2\times$. FLASHCONV trains H3 4-8$\times$ times faster than attention for long sequences, and is a critical component for scaling to billion-parameter models[2].

---

[1]An A100 GPU has a maximum of 312 TFLOPs/s of FP16 with tensor cores, but only 20 TFLOPs/s of FP32 (and 40 TFLOPs/s of FP16) without tensor cores (NVIDIA, 2020). This trend started with the V100 GPUs (NVIDIA, 2017) and has continued with the H100 GPUs (NVIDIA, 2022b).

[2]Code for H3 is available at https://github.com/HazyResearch/H3.

## 2 BACKGROUND

We present some background on state space models and linear attention, which inspired our H3 layer.

### 2.1 STATE SPACE MODELS

A continuous-time state-space representation (Brogan, 1974) defines a linear mapping from an input signal $u(t) \in \mathbb{R}$ (as a function of time $t$) to an output signal $y(t) \in \mathbb{R}$ through a state-variable $x(t) \in \mathbb{R}^m$, with the following differential equation, for some matrices $\mathbf{A} \in \mathbb{R}^{m \times m}$, $\mathbf{B} \in \mathbb{R}^{m \times 1}$, $\mathbf{C} \in \mathbb{R}^{1 \times m}$, $\mathbf{D} \in \mathbb{R}^{1 \times 1}$: $\dot{x}(t) = \mathbf{A}x(t) + \mathbf{B}u(t), y(t) = \mathbf{C}x(t) + \mathbf{D}u(t)$.

Similarly, a discrete-time state-space representation defines a linear mapping from a discrete input signal $u_i$ (for $i = 1, 2, ...$) to a discrete output signal $y_i$ though a state-variable $x_i \in \mathbb{R}^m$:

$$x_i = \mathbf{A}x_{i-1} + \mathbf{B}u_i$$
$$y_i = \mathbf{C}x_i + \mathbf{D}u_i.$$

A state-space model (SSM) uses these representations as a layer in a deep learning pipeline, where the matrices $\mathbf{A}, \mathbf{B}, \mathbf{C}, \mathbf{D}$ are learned from data (e.g., with gradient-based optimization). One often has $d$ of these SSMs in parallel, each corresponding to one hidden dimension. To preserve the sequence history, HiPPO (Gu et al., 2020) projects the history on a basis of orthogonal polynomials, which translates to having SSMs whose $\mathbf{A}, \mathbf{B}$ matrices are initialized to some special matrices.

This recurrent form of SSMs allows efficient inference (i.e., generation): to generate the output of the next time-step, one only needs the state of the current time-step, not the entire input history. Furthermore, SSMs can freely extrapolate to sequences longer than seen during training.

**SSMs as Convolution.** For efficient training, given the entire sequence of the input $u_1, ..., u_N$, the output sequence $y_1, ..., y_N$ can also be written as the convolution of the input with the filter (Gu et al., 2021):

$$f = [\mathbf{CB}, \mathbf{CAB}, \mathbf{CA}^2\mathbf{B}, ..., \mathbf{CA}^{N-1}\mathbf{B}].$$

That is, from an initial condition $x_0$, we have $y_i = \mathbf{CA}^i\mathbf{B}x_0 + (f * u)_i + \mathbf{D}u_i$, where $(f * u)$ denotes a linear convolution between $f$ and $u$. If we set the initial condition $x_0$ to be zero, then $y$ is exactly a linear convolution of $u$, with a residual connection $\mathbf{D}u$. More generally, any linear time-invariant system (of which SSMs are a special case) can be written as a convolution.

Given a 1D input sequence $u \in \mathbb{R}^N$ of length $N$, we denote the 1D output sequence $y \in \mathbb{R}^N$ of an SSM parameterized by matrices $\mathbf{A}, \mathbf{B}, \mathbf{C}, \mathbf{D}$ as

$$y = \mathrm{SSM}_{\mathbf{A}, \mathbf{B}, \mathbf{C}, \mathbf{D}}(u).$$

To simplify notation, we omit the reference to $\mathbf{A}, \mathbf{B}, \mathbf{C}, \mathbf{D}$ and write $y = \mathrm{SSM}(u)$ if they are clear from context. When $u$ is multidimensional of dimension $d$, we stack $d$ of these SSMs together that defines a mapping from $u \in \mathbb{R}^{N \times d}$ to $y \in \mathbb{R}^{N \times d}$, using the same notation $y = \mathrm{SSM}(u)$.

To construct the filter $f$ from $\mathbf{A}, \mathbf{B}, \mathbf{C}$ efficiently, $\mathbf{A}$ is often constrained to be diagonal (Gupta et al., 2022; Gu et al., 2022b), or diagonal plus low-rank (Gu et al., 2022a).

**SSM through FFTs.** Computing the convolution naively through conventional matrix operations is expensive for long kernels, scaling as $O(N^2)$. Instead, we can use FFTs: take the FFT of $f$ and $u$, multiply them together pointwise, and then take the inverse FFT. This yields an $O(N \log N)$ algorithm.

### 2.2 LINEAR ATTENTION

We describe linear attention (Katharopoulos et al., 2020) and its connection to RNNs, which inspired our model design (Section 3).

In standard attention (Vaswani et al., 2017), we have $N$ query/key/value tokens $Q_i, K_i, V_i \in \mathbb{R}^d$ for $i = 1, ..., N$, where $N$ is the sequence length and $d$ is the head dimension. For some similarity metric $\mathrm{Sim} : \mathbb{R}^d \times \mathbb{R}^d \to \mathbb{R}$, we want to compute the output:

$$O_i = \frac{\sum_{j=1}^i \mathrm{Sim}(Q_i, K_j)V_j}{\sum_{j=1}^i \mathrm{Sim}(Q_i, K_j)} \in \mathbb{R}^d.$$

For standard softmax attention, $\mathrm{Sim}(q, k) = e^{q^\top k}$ (often the dot product is scaled by $1/\sqrt{d}$). Linear attention makes the assumption that $\mathrm{Sim}$ has the form $\mathrm{Sim}(q, k) = \phi(q)^\top \phi(k)$, for some (nonlinear) function $\phi$. The output is then $O_i = \frac{\phi(Q_i)^\top \sum_{j=1}^i \phi(K_j)V_j^\top}{\phi(Q_i)^\top \sum_{j=1}^i \phi(K_j)}$. Let $S_i = \sum_{j=1}^i \phi(K_j)V_j^\top \in \mathbb{R}^{d \times d}$, $z_i = \sum_{j=1}^i \phi(K_j) \in \mathbb{R}^d$, $d_i = \phi(Q_i)^\top z_i \in \mathbb{R}$. Then $O_i = \frac{\phi(Q_i)^\top S_i}{d_i}$. This connects linear attention to RNNs: the output $O_i$ is a function of $S_i$ and $z_i$, both of which are incrementally updated (as cumulative sums).

## 3 HUNGRY HUNGRY HIPPOS LAYER TO MODEL DISCRETE SEQUENCES

To understand the gap between SSMs and attention on language modeling, we examine two synthetic language modeling tasks. These tasks motivate our H3 layer to add a discrete SSM (based on shift matrix) and multiplicative interaction to effectively model discrete sequences. We then show that the H3 layer is expressive enough to solve these synthetic tasks, and that this understanding leads to better performance on a real language modeling benchmark.

### 3.1 MOTIVATION: SYNTHETIC LANGUAGE MODELING TASKS

We describe two closely-related synthetic tasks, summarized in Table 1. Olsson et al. (Olsson et al., 2022) argue that the ability to solve (variants of) these tasks accounts for the majority of the in-context learning capability of Transformers, and more intuition is given in Appendix E.

Table 1: Synthetic language modeling tasks.

| Task | Input | Output | Sequence Length | Vocab Size |
|---|---|---|---|---|
| Induction Head | $a\,b\,c\,d\,e \vdash f\,g\,h\,i\,...\,x\,y\,z \vdash$ | $f$ | 30 | 20 |
| Associative Recall | $a\,2\,c\,4\,b\,3\,d\,1\,a$ | 2 | 20 | 10 |

The **Induction Head** task tests how well a model can recall content after a special token (e.g., $\vdash$ in Table 1). At the end of the sequence, the model must recall the token that appeared immediately after the special token earlier in the sequence. **Associative Recall** (Ba et al., 2016) is similar to the induction head task, but requires the model to remember multiple key-value pairs. At the end of the sequence, the model must recall a specific value belonging to a specific key.

Table 2: Evaluation of 2-layer models on synthetic language tasks.

| Task | Random | S4D | Gated State Spaces | H3 | Attention |
|---|---|---|---|---|---|
| Induction Head | 5.0 | 35.6 | 6.8 | **100.0** | **100.0** |
| Associative Recall | 25.0 | 86.0 | 78.0 | 99.8 | **100.0** |

Table 2 (for two-layer models) shows that S4D (Gu et al., 2022b) and Gated State Spaces (Mehta et al., 2022) both fail to model these synthetic languages, which suggests they may not have the expressivity for general language. We argue that these failures suggest two missing capabilities: (i) to remember tokens that appear after a particular event (e.g., the special token in the induction head task), and (ii) to compare tokens across the sequence (e.g., comparing keys to decide which value to recall). Attention has both these capabilities: it can compare tokens by constructing the *quadratic* attention matrix $\mathbf{Q}\mathbf{K}^\top$, and it can recall tokens by direct copying (multiplying $\mathrm{softmax}(\mathbf{Q}\mathbf{K}^\top)$ with $\mathbf{V}$). In Section 3.2, we design our new layer H3 to enable these capabilities in SSMs, narrowing the expressivity gap between SSMs and attention.

### 3.2 H3 LAYER

H3 uses SSMs with shift and diagonal matrices, along with multiplicative operations against projections of the input to capture the missing capabilities identified by the synthetics.

**High-level Intuition.** (i) To remember tokens from the past, we want the state $x_i$ to copy from the input $u_i$, and then pass that information to the next state $x_{i+1}$. As $x_{i+1}$ relates to $x_i$ by $\mathbf{A}x_i$, we use a discrete SSM with a shift matrix $\mathbf{A}$ (described formally below) that shifts the elements of a state vector (e.g., mapping $[a,b,c] \to [0,a,b]$). (ii) To compare tokens across the sequence, we use multiplicative interaction: the output of an SSM, containing information from previous time steps, is multiplied with the input at the current time steps, thus measuring similarity between tokens.

H3 is loosely inspired by linear attention (Section 2): we project the input $u$ to get three signals $\mathbf{Q},\mathbf{K},\mathbf{V}$. Then we replace the non-linearity $\phi(\mathbf{K})$ with an SSM where $\mathbf{A}$ is a shift matrix ($\mathrm{SSM}_{\mathrm{shift}}$), and we replace the summation $S_i$ with a SSM with diagonal $\mathbf{A}$ ($\mathrm{SSM}_{\mathrm{diag}}$). The output, for the case of head dimension $d_h = 1$, is:

$$\mathbf{Q} \odot \mathrm{SSM}_{\mathrm{diag}}(\mathrm{SSM}_{\mathrm{shift}}(\mathbf{K}) \odot \mathbf{V}),$$

where $\odot$ denotes pointwise multiplication. We can view this form as stacking two SSMs with multiplicative interaction (each is a "hungry hippo", hence the name of our layer). A more formal connection between linear attention, time-varying systems, and H3 can be found in Appendix B.

**Remembering Key Tokens: Shift and Diagonal SSMs.** The shift and diagonal SSMs are designed to address the capability to log tokens after particular events. In the shift SSM, we constrain $\mathbf{A} \in \mathbb{R}^{m \times m}$ to be a shift matrix $\mathbf{A}_{i,j} = \begin{cases} 1 & \text{for } i-1=j \\ 0 & \text{otherwise} \end{cases}$. The action of this matrix on the hidden state $x_i$ is to shift each coordinate down by one—thereby creating a "memory" of the previous states. For example, if $\mathbf{B} = e_1$, the first

basis vector, then $x_i = [u_i, u_{i-1}, ..., u_{i-m+1}]$ contains the inputs from the previous $m$ time steps. We learn $\mathbf{B}$ and $\mathbf{C}$ ($\mathbf{B}$ can also be fixed to $e_1$ for simplicity, in which case the output is a 1D conv. with kernel size $m$).

The diagonal SSM constrains $\mathbf{A}$ to be diagonal and initializes it from the diagonal version of HiPPO (S4D (Gu et al., 2022b)). This parameterization allows the model to remember state over the entire sequence. The shift SSM can detect when a particular event occurs, and the diagonal SSM can remember a token afterwards for the rest of the sequence.

**Multiplicative Interaction for Comparison.** We take the multiplicative interactions from linear attention, but they provide another missing capability when combined with a shift matrix: comparing tokens across the sequence. The multiplicative interactions between the output of the shift SSM and the $\mathbf{V}$ projection mimics local multiplicative interactions in linear attention (depending on the size of the hidden state). Similarly, multiplicative interactions with the $\mathbf{Q}$ projection and the output of the diagonal SSM allows comparisons between tokens over the entire sequence.

**H3 Layer.** The overall layer is given in Algorithm 1 and shown schematically in Figure 1 (left). We use the H3 layer to construct a model in the same style as Transformers by interleaving it with MLPs, connected by residual connection and layer norm (i.e., pre-norm architecture (Baevski & Auli, 2018)). We will also consider a hybrid H3-attention model (two attention layers while the rest are H3, Sections 3.3 and 5).

---

**Algorithm 1** H3 Layer

---

**Require:** Input sequence $u \in \mathbb{R}^{N \times d}$ from the previous layer, weight matrices $\mathbf{W}_Q, \mathbf{W}_K, \mathbf{W}_V, \mathbf{W}_O \in \mathbb{R}^{d \times d}$, a shift SSM $\mathrm{SSM}_{\mathrm{shift}}$, a diagonal SSM $\mathrm{SSM}_{\mathrm{diag}}$, head dimension $d_h$.

1: Compute $\mathbf{Q} = u\mathbf{W}_Q, \mathbf{K} = u\mathbf{W}_K, \mathbf{V} = u\mathbf{W}_V \in \mathbb{R}^{N \times d}$.
2: Pass $\mathbf{K}$ through the shift SSM: $\overline{\mathbf{K}} = \mathrm{SSM}_{\mathrm{shift}}(\mathbf{K}) \in \mathbb{R}^{N \times d}$.
3: Split $\mathbf{Q}, \overline{\mathbf{K}}, \mathbf{V}$ into $H$ "heads" ($\mathbf{Q}^{(h)}, \overline{\mathbf{K}}^{(h)}, \mathbf{V}^{(h)}$ for $h = 1, ..., H$), each a sequence of $N$ vectors of size $d_h = d/H$.
4: **for** $1 \le h \le H$ **do**
5:    Take the batched outer product $\overline{\mathbf{K}}^{(h)}(\mathbf{V}^{(h)})^\top \in \mathbb{R}^{N \times d_h \times d_h}$ (batched in the $N$-dimension) and pass it through a diagonal SSM: $\mathbf{KV}^{(h)} = \mathrm{SSM}_{\mathrm{diag}}(\overline{\mathbf{K}}^{(h)}(\mathbf{V}^{(h)})^\top) \in \mathbb{R}^{N \times d_h \times d_h}$.
6:    Batch-multiply by $\mathbf{Q}$: $\mathbf{O}^{(h)} = [\mathbf{Q}_1^{(h)}\mathbf{KV}_1^{(h)}, ..., \mathbf{Q}_N^{(h)}\mathbf{KV}_N^{(h)}] \in \mathbb{R}^{N \times d_h}$ (batched in the $N$-dimension).
7: **end for**
8: Concatenate the output $\mathbf{O}^{(h)}$ of each head, and multiply by the output projection matrix $\mathbf{W}_O \in \mathbb{R}^{d \times d}$.

---

**Efficiency** We show that H3 scales as $O(N \log N)$ with sequence length $N$—asymptotically more efficient than attention, which typically requires $O(N^2 d)$ time and $O(N^2)$ space[3] (proof in Appendix D.3).

**Proposition 1.** *Let $N$ be the sequence length, $d$ be the hidden dimension, and assume that the head dimension $d_h$ is of order $O(1)$. Then the H3 layer takes $O(d^2 N + dN \log N)$ time and $O(dN)$ space to compute.*

## 3.3 EXPRESSIVITY

We show that H3 can model our synthetic languages, as well as natural language on OpenWebText (Gokaslan et al., 2019). We also present a hybrid H3-attention extension that outperforms Transformers on OpenWebText.

**Mechanism for Solving Associative Recall with H3.** H3 is expressive enough to solve our synthetic language modeling tasks, as shown in Table 2. Figure 1 (middle) shows a mechanism for a single H3 layer to solve the associative recall task for a particular key-value pair $(a, 3)$. The shift SSM and following multiplicative interaction act as a gate on whether to let a value through to the diagonal SSM, based on whether the previous token was key $a$. The diagonal SSM stores the value 3 in memory, and continually outputs it. The final multiplicative interaction gates whether to let the diagonal SSM's output through—based on whether the current input token is the key $a$. We formally construct the weights of an H3 layer to solve this task in Appendix D.1.

Table 3: Perplexity of SSM variants compared to Transformers on OpenWebText. All models have 12 layers, with size around 125M, and are trained with the same hyperpameters, for 50B tokens.

| H3 | H3 Hybrid (2 Attn) | S4D | GSS | GSS Hybrid (2 Attn) | Transformer |
|----|--------------------|-----|-----|---------------------|-------------|
| 21.0 | **19.6** | 24.9 | 24.0 | 19.8 | 20.6 |

**Better Synthetic Language Modeling Translates to Better Natural Language Modeling.** We validate that when H3 can solve these synthetic tasks, it also improves the modeling capability on natural language

---

[3]There are several memory-efficient algorithms for attention (Rabe & Staats, 2021; Dao et al., 2022b), though their time complexity is still quadratic in $N$, which is a lower-bound for attention (Keles et al., 2022).

(e.g., on the OpenWebText dataset). As shown in Table 3, H3 comes within 0.4 perplexity points of Transformers when trained for 50B tokens on OpenWebText, and performs much better than existing SSM variants (S4D, GSS), by $3-3.9$ points.

**Extension: H3-attention Hybrid Model.** A simple hybrid H3-attention language model surprisingly outperforms Transformers on OpenWebText by 1.0 point. Our hybrid model simply retains two self-attention layers: one in the second layer, and one in the middle (layer $2+N/2$ for an $N$-layer model, $N$ even). The H3-attention hybrid also outperforms the GSS-attention hybrid (Mehta et al., 2022).

## 4 FLASHCONV: EFFICIENTLY TRAINING SSMs

To improve the efficiency of SSMs on modern hardware, we propose FLASHCONV. FLASHCONV fuses the FFT, pointwise multiply, and inverse FFT to reduce memory reads/writes. It also uses a block FFT algorithm to make use of specialized matrix multiply units (e.g., tensor cores on A100) for sequence lengths up to 8K. For sequences longer than 8K, the computation no longer fits in GPU SRAM[4], so we propose a novel state-passing algorithm that splits the sequence into chunks to compute the FFT convolution one chunk at a time. FLASHCONV can speed up any SSMs (not just H3).

### 4.1 FUSED BLOCK FFTCONV

We deploy two techniques to speed up the FFT-based convolution for sequences shorter than 8K: kernel fusion and block FFT. Kernel fusion addresses IO bottlenecks due to reading and writing of intermediate results, while block FFT allows the FFT-based convolution to utilize specialized matrix multiplication units. These techniques allow us to speed up FFTConv by $2\times$ (Section 6) for sequences shorter than 8k.

**Kernel Fusion.** Naive implementations of FFTConv using standard libraries such as cuFFT are IO-bound due to repeated reading and writing of intermediate results. The FFT convolution in an SSM with input $u$ and filter $f$ has the form $iFFT(FFT(u) \odot FFT(f))$ (where $\odot$ denotes pointwise multiplication). It requires reading and writing intermediate results to GPU memory—which can dominate the runtime. Following FLASHATTENTION (Dao et al., 2022b), we first fuse the entire FFTConv into a single kernel and compute it in SRAM to avoid this overhead.

**Block FFT.** To further speed up the computation of FFT-based convolution, we exploit specialized matrix multiplication hardware on modern GPUs (e.g., Tensor Cores on Nvidia GPUs perform fast $16 \times 16$ matrix multiplication). We appeal to classical results that show that the FFT can be written as a series of block-diagonal matrix multiplications interleaved with permutation. We note that such algorithms are not new, but our setting (fused FFTConv on GPU) introduces new bottlenecks—by removing the IO bottlenecks, compute becomes the bottleneck (note that a single FFT on GPU is usually IO bound).

Suppose that we want to perform an $N$-point FFT, which is equivalent to multiply by the DFT matrix $\mathbf{F}_N$. Suppose that $N = N_1 N_2$ for some integers $N_1, N_2$. By the Cooley-Tukey decomposition of the DFT (Cooley & Tukey, 1965; Bailey, 1990) (also known as the four-step FFT algorithm), we can write $\mathbf{F}_N = \mathbf{P}(\mathbf{I}_{N_2} \otimes \mathbf{F}_{N_1})\mathbf{P}^\top \mathbf{D}(\mathbf{I}_{N_1} \otimes \mathbf{F}_{N_2})\mathbf{P}$, where $\mathbf{P}$ denotes a fixed permutation that reshapes the input as a $N_1 \times N_2$ array and then transpose it, $\otimes$ denotes Kroneker product, $\mathbf{D}$ is a $N \times N$ diagonal matrix (called the twiddle factors) (Dao et al., 2022a), and $\mathbf{I}_{N_i}$ and $\mathbf{F}_{N_i}$ are the identity and DFT matrix of size $N_i \times N_i$. As $\mathbf{I}_{N_2} \otimes \mathbf{F}_{N_1}$ and $\mathbf{I}_{N_1} \otimes \mathbf{F}_{N_2}$ are just block-diagonal matrices, we can make use of specialized matmul units to perform these multiplications. Similarly, if $N = N_1 N_2 N_3$ then we can decompose the $N$-point FFT into a series of (block) FFT of size $N_1$, $N_2$, and $N_3$, interleaved by permutation.

The block FFT algorithm incurs $O(Nr \log N / \log r)$ FLOPs for a sequence length $N$, if $N$ can be written as $r^p$ for two integers $r, p$. This incurs more FLOPs than standard FFT ($O(N \log N)$), but can run faster when we using specialized matrix multiplication hardware.

### 4.2 STATE-PASSING

However, the fused kernel cannot run if the sequence is too long to fit into GPU SRAM (longer than 8K on A100). We show how to exploit the particular form of the FFT in SSM to speed it up for long sequences.

The recurrent nature of SSMs allows us to split the FFTConv of a length-$N$ sequence into chunks of size $N'$ each ($N'$ is the longest FFT we can fit into SRAM), assuming $N$ is a multiple of $N'$. We use FFTConv to compute each chunk and use a recurrence to connect the chunks. In particular, we split the inputs $u$ into $C = N/N'$ chunks $u^{(c)} \in \mathbb{R}^{N'}$ for $c = 1, ..., C$. Similarly, split the states $x$ into $x^{(c)} \in \mathbb{R}^{N' \times m}$ and the output $y$ into $y^{(c)} \in \mathbb{R}^{N'}$ for $i = 1, ..., C$. We will only need the end-state $x_{N'}^{(c)}$ of each chunk $c$.

---

[4]SRAM, or on-chip memory, is much faster than off-chip GPU memory, but usually much smaller, on the order of around 100KB for each streaming processor.

Table 4: Perplexity (lower is better) of models on the Pile, OpenWebText and WikiText-103. GPT-Neo and hybrid H3 are trained on the Pile, while GPT2 is trained on WebText. All models use the same GPT2 tokenizer. We report the perplexity of GPT-2 models on the Pile ($^*$) for context, though the performance is not directly comparable since they were trained on different data.

| Model | Pile | OpenWebText | WikiText103 |
|---|---|---|---|
| GPT-2 small (125M) | 19.0* | 22.6 | 29.9 |
| GPT-Neo-125M | 9.4 | 22.6 | 26.3 |
| **Hybrid H3-125M** | **8.8** | **20.9** | **23.7** |
| GPT-2 medium (355M) | 13.9* | 17.0 | 21.8 |
| **Hybrid H3-355M** | **7.1** | **15.9** | **16.9** |
| GPT-2 XL (1.5B) | 12.4* | 12.9 | 17.0 |
| GPT-Neo-1.3B | 6.2 | 13.1 | 13.3 |
| **Hybrid H3-1.3B** | **6.0** | **12.4** | **12.5** |
| GPT-Neo-2.7B | 5.7 | 11.7 | 11.5 |
| **Hybrid H3-2.7B** | **5.4** | **11.0** | **10.6** |

Let $f = [\mathbf{CB}, \mathbf{CAB}, \mathbf{CA}^2\mathbf{B}, ..., \mathbf{CA}^{N'-1}\mathbf{B}]$ be the SSM filter. Recall from Section 2 that for each chunk $c$, $y_i^{(c)} = \mathbf{CA}^i\mathbf{B}x_{N'}^{(c-1)} + (f * u^{(c)})_i + \mathbf{D}u_i^{(c)}$, since $x_{N'}^{(c-1)}$, the end-state of the previous chunk $(c-1)$ is the initial condition for the current chunk $c$. In vector notation, $y^{(c)} = \mathbf{M}_{xy}x_{N'}^{(c-1)} + f * u^{(c)} + \mathbf{D}u^{(c)}$ for some matrix $\mathbf{M}_{xy} \in \mathbb{R}^{N' \times m}$. Additionally we need to update the end-state of each chunk with $x_{N'}^c = \mathbf{A}^{N'}x_{N'}^{(c-1)} + \mathbf{M}_{ux}u^{(c)}$ for some matrix $\mathbf{M}_{ux}^{m \times N'}$ (derivation in Appendix C.2). In essence, we can compute the output for each chunk with FFT-based convolution as long as we remember the end-state of the previous chunk, and the end-state of each chunk can be updated recurrently. This yields a state-passing algorithm for long sequences, where we only compute FFT of length $N'$, and update some hidden state each iteration.

Let BLOCKFFTCONV refer to our fused block FFTConv kernel. Then, the state-passing algorithm for 1D input is given by Algorithm 2. For inputs of dimension $d$ where we stack $d$ SSMs, we simply batch Algorithm 2 along the $d$-dimension.

---

**Algorithm 2** State Passing Algorithm

---

**Require:** Input $u \in \mathbb{R}^N$, SSM parameterized by matrices $\mathbf{A} \in \mathbb{R}^{m \times m}$, $\mathbf{B} \in \mathbb{R}^{m \times 1}$, $\mathbf{C} \in \mathbb{R}^{1 \times m}$, $\mathbf{D} \in \mathbb{R}^{1 \times 1}$, chunk size $N'$ where $N$ is a multiple of $N'$.
1: Precompute $\mathbf{A}^{N'} \in \mathbb{R}^{m \times m}$, $\mathbf{M}_{ux} = [\mathbf{A}^{N'-1}\mathbf{B}, ..., \mathbf{B}] \in \mathbb{R}^{m \times N'}$, $\mathbf{M}_{xy} = [\mathbf{C}, \mathbf{CA}, ..., \mathbf{CA}^{N'-1}] \in \mathbb{R}^{N' \times m}$.
2: Split the inputs $u_{1:N}$ into $C = N/N'$ chunks $u_{1:N'}^{(c)}$ for $c = 1, ..., C$.
3: Let the initial state be $x_{N'}^{(0)} = 0 \in \mathbb{R}^m$.
4: **for** $1 \leq c \leq C$ **do**
5:    Compute $y^{(c)} = \mathbf{M}_{xy}x_{N'}^{(c-1)} + \text{BLOCKFFTCONV}(f, u_j) + \mathbf{D}u^{(c)} \in \mathbb{R}^{N'}$.
6:    Update state: $x_{N'}^{(c)} = \mathbf{A}^{N'}x_{N'}^{(c-1)} + \mathbf{M}_{ux}u^{(c)}$.
7: **end for**
8: Return $y = [y^{(1)}...y^{(C)}]$.

---

We prove that Algorithm 2 yields the same output as if one has computed the SSM using a large FFT of size $N$ (proof in Appendix D.4):

**Proposition 2.** *For input $u \in \mathbb{R}^N$ and matrices $\mathbf{A}, \mathbf{B}, \mathbf{C}, \mathbf{D}$, the output $y \in \mathbb{R}^N$ returned by Algorithm 2 is the same as the output defined by the SSM parameterized by $\mathbf{A}, \mathbf{B}, \mathbf{C}, \mathbf{D}$.*

## 5 H3 EVALUATION

To understand how well capturing the synthetics in Section 3.1 translates to language modeling, we train two hybrid hybrid H3-attention language models at sizes 125M, 355M, 1.3B, and 2.7B, and we evaluate their performance against Transformers. The hybrid models match or exceed the quality of Transformers in perplexity and zero/few-shot learning. We also validate that H3 models retain strong performance on non-text sequence modeling. Appendix F contains additional experiments on more datasets, length extrapolation, and scaling with data.

### 5.1 LANGUAGE MODELING

We compare hybrid H3-attention language models against Transformer-based language models. We evaluate language modeling performance using perplexity, zero-shot learning, and few-shot learning performance. Hybrid H3 models outperform Transformers, which suggests that closing the gap between

Table 5: Zero-shot acc. on SuperGLUE with logit scoring. Best results in bold, second best underline.

| Model | WSC | WIC | RTE | CB | MultiRC | ReCoRD | BoolQ | COPA | Average |
|---|---|---|---|---|---|---|---|---|---|
| OPT-125M | **39.4** | 52.0 | 48.7 | 37.4 | 58.9 | 44.9 | 59.6 | 60.0 | 50.1 |
| GPT-Neo-125M | 36.5 | **53.6** | 53.1 | 41.1 | **59.9** | 39.6 | **62.2** | 60.0 | 50.8 |
| **Hybrid H3-125M** | **39.4** | 51.4 | **59.2** | **48.2** | 51.4 | **55.0** | 59.6 | **67.0** | **53.9** |
| GPT-2 medium (355M) | 50.0 | 52.0 | 51.3 | 28.6 | **59.5** | 53.3 | 61.0 | 65.0 | 52.6 |
| OPT-350M | **53.5** | 50.8 | 53.4 | 35.7 | 58.9 | 51.4 | 60.9 | 60.0 | 53.1 |
| **Hybrid H3-355M** | 37.5 | 51.7 | **55.2** | **41.1** | 59.5 | **62.3** | **61.5** | **69.0** | **54.7** |
| OPT-1.3B | 36.5 | 49.5 | **53.4** | 39.3 | **58.3** | 61.8 | 55.0 | 69.0 | 52.9 |
| GPT-Neo-1.3B | 41.3 | 50.0 | 52.3 | 33.9 | 57.9 | 55.5 | 59.9 | 66.0 | 52.1 |
| **Hybrid H3-1.3B** | 52.9 | 50.3 | 53.4 | 33.9 | 58.2 | 67.8 | 61.7 | 74.0 | **56.5** |
| OPT-2.7B | **51.0** | 50.8 | 50.5 | 41.1 | 57.4 | 65.9 | 60.9 | 66.0 | 55.5 |
| GPT-Neo-2.7B | 37.5 | 50.0 | 52.3 | 50.0 | **59.1** | 60.0 | **61.1** | 67.0 | 54.6 |
| **Hybrid H3-2.7B** | 36.5 | 51.3 | **57.0** | 37.5 | 58.7 | **71.3** | 61.1 | **81.0** | **56.8** |

Table 6: 3-shot acc. on SuperGLUE with logit scoring. Best results in bold, second best underline.

| Model | WSC | WIC | RTE | CB | MultiRC | ReCoRD | BoolQ | COPA | Average |
|---|---|---|---|---|---|---|---|---|---|
| OPT-125M | 36.5 | **50.2** | 47.3 | 44.6 | 57.9 | 44.9 | 41.9 | 60.0 | 47.9 |
| GPT-Neo-125M | 38.5 | 50.0 | 53.1 | 17.9 | 56.3 | 39.6 | **62.1** | 60.0 | 47.2 |
| **Hybrid H3-125M** | **43.3** | 49.1 | **58.1** | **51.8** | 48.9 | **55.0** | 56.1 | **67.0** | **53.7** |
| GPT-2 medium (355M) | 36.5 | **50.5** | 48.0 | 8.9 | 43.5 | 53.3 | 58.8 | 65.0 | 45.6 |
| OPT-350M | 37.5 | 50.0 | 45.8 | **44.6** | 49.8 | 51.4 | **61.7** | 60.0 | 50.1 |
| **Hybrid H3-355M** | **42.3** | 47.5 | **50.5** | 28.6 | **59.7** | **62.3** | 60.5 | **69.0** | **52.6** |
| OPT-1.3B | **44.2** | 51.1 | 53.4 | 16.1 | **59.9** | 62.1 | 38.3 | 70.0 | 49.4 |
| GPT-Neo-1.3B | 35.6 | 50.6 | 47.3 | **32.1** | **59.9** | 55.7 | 61.2 | 67.0 | 51.2 |
| **Hybrid H3-1.3B** | 36.5 | 49.2 | **55.2** | 23.2 | 59.3 | **67.6** | 56.9 | **76.0** | **53.0** |
| OPT-2.7B | 44.2 | 50.5 | **53.4** | 17.9 | 59.2 | 66.0 | **62.0** | 71.0 | 53.0 |
| GPT-Neo-2.7B | **49.0** | 51.9 | 51.6 | 21.4 | 57.0 | 60.0 | 56.0 | 68.0 | 51.9 |
| **Hybrid H3-2.7B** | 36.5 | 45.6 | 47.3 | **46.4** | **59.4** | **71.1** | 60.6 | **77.0** | **55.5** |

Table 7: Inference throughput on A100 80GB, 1.3B models. Batch size 64, prompt length 512, 1024, or 1536, and generating 128 tokens per sequence in the batch (i.e., $64 \times 128$ tokens in a batch). Hybrid H3 is up to $2.4\times$ faster than a Transformer of similar size in inference. The difference is larger for longer sequences.

| Tokens/s | Prompt length 512 | Prompt length 1024 | Prompt length 1536 |
|---|---|---|---|
| Transformer-1.3B | 1340 | 770 | 520 |
| Hybrid H3-1.3B | 1980 | 1580 | 1240 |

SSMs and attention on the synthetic languages translates to real language modeling capabilities. We also report the generation speed of hybrid H3 models compared to Transformers; since SSMs are recurrent models, they can generate tokens $2.4\times$ faster than Transformers. Appendix F shows performance of pure H3 language models on these same evaluation metrics.

**Setup**   We train hybrid models at sizes 125M, 355M, 1.3B, and 2.7B on the Pile (Gao et al., 2020) for 400B tokens. We compare against checkpoints of equivalent sizes from Open-AI (Radford et al., 2019) and GPT-Neo[5] (Black et al., 2021), from HuggingFace (Wolf et al., 2020).

**Perplexity**   Table 4 shows perplexity on the Pile (Gao et al., 2020), OpenWebText (Gokaslan et al., 2019), and WikiText-103 (Merity et al., 2016). On the Pile, our 125M hybrid model outperforms GPT-Neo, which was also trained on the Pile. Our hybrid models also outperform GPT-Neo models and GPT-2 models on zero-shot transfer to OpenWebText and WikiText103. We report the PPL of GPT-2 models for context, though the performance is not directly comparable since they were trained on different data.

**Zero- and Few-shot Performance**   We compare the zero- and few-shot performance of hybrid H3 language models against OPT (Zhang et al., 2022), GPT-Neo, and GPT-2 models, where public checkpoints are available. We report performance with rank classification on the logits of the possible choices (see Appendix F.7 for raw generation). Table 5 reports zero-shot performance on the SuperGLUE benchmark, and Table 6 reports the 3-shot performance. Following the perplexity results, the hybrid language models outperform or match the best the Transformer baseline on more than half the tasks.

**Language Modeling Inference**   Finally, since SSMs are recurrent models, they admit faster text generation than Transformers. Table 7 shows inference throughput of a 1.3B-parameter hybrid model compared to a Transformer. The hybrid model has up to $2.4\times$ higher throughput.

# 6 FLASHCONV EVALUATION

We evaluate how well FLASHCONV speeds up SSMs. FLASHCONV sets state-of-the-art performance on the long range arena benchmark (Tay et al., 2020) using S4 (Gu et al., 2022a). We report performance

---

[5]There is no pretrained GPT-Neo at the 350M size.

Table 8: Speedup on the LRA benchmark.

| Models | Speedup |
|---|---|
| Transformer | 1× |
| FlashAttention (Dao et al., 2022b) | 2.4× |
| Block-sparse FlashAttention (Dao et al., 2022b) | 2.8× |
| S4 (Gu et al., 2022c) | 2.9× |
| S4 with FLASHCONV | 5.8× |

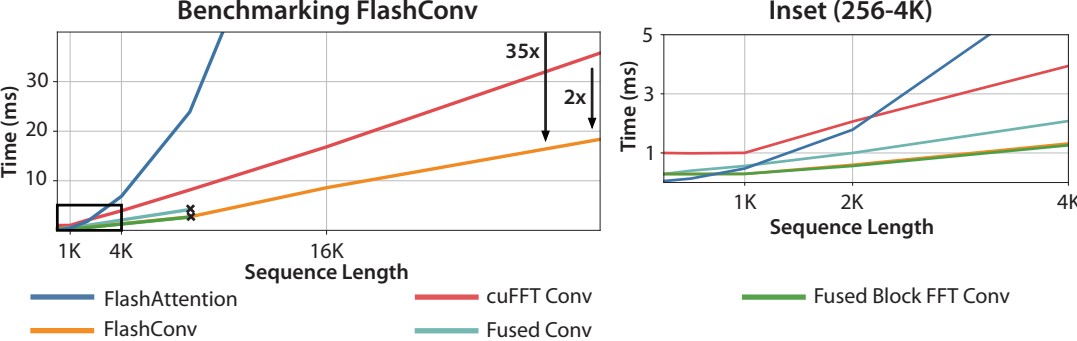

Figure 2: We compare the speed of different algorithms to perform FFT-based convolution, along with FlashAttention (Dao et al., 2022b) (the fastest attention implementation we know of). We use batch size 8, hidden dimension 1024, and varying sequence length from 256 to 32k, and measure on an A100-SMX4-40GB GPU. We see that kernel fusion gives up to 3.4× speedup over naive FFTConv for short sequences (up to 512), block FFT gives up to 2× speedup for medium length sequences (1k - 8k), and state-passing allows 2.3× faster FFTConv for long sequences (16k and above).

of training H3 module with FLASHCONV compared to attention at various sequence lengths, from 256 to 32K and demonstrate nearly linear scaling.

**Long Range Arena**    The Long Range Arena (LRA) benchmark (Tay et al., 2020) is a benchmark for long-range sequence modeling. The state-of-the-art approach, S4 (Gu et al., 2022c), is an SSM. Table 8 shows that FLASHCONV accelerates S4 by 2×, outperforming Transformers by 5.8×.

**Benchmark H3 Against Attention**    We benchmark the time to run the forward and backward pass of H3 with FLASHCONV against attention. FLASHCONV maintains nearly linear scaling, even to very long sequence lengths. Fig. 2 shows overall 2-3× speedup over FFTConv with cuFFT using our techniques (block FFT, state-passing). Simple kernel fusion (even without block FFT) can yield speedup over cuFFT for short sequences, since memory reads/writes are the bottleneck for short sequences. For long sequences, SSMs using state passing can be dozens of times faster than even the fastest attention implementation.

## 7    CONCLUSION

Our main goal is to understand and narrow the gap between attention and SSMs in language modeling in terms of modeling capabilities and hardware efficiency. Our exploration based on synthetic language tasks motivated us to design the H3 layer, which is surprisingly competitive with attention. Our BLOCKFFTCONV algorithm exploits matrix multiplication units and the dual recurrent–convolution view of SSMs to substantially speed up SSMs, reducing the hardware barrier between attention and SSMs. We are excited about several future directions. Our H3 layer is a simple combination of two SSMs, and more sophisticated designs could be more expressive. Our encouraging results on language models up to 1.3B parameters suggests that scaling SSMs to larger sizes is a promising avenue. Since simply adding two attention layers to H3 models already outperforms both the pure H3 model and Transformers, we are optimistic about combining the complementary strengths of SSMs and attention in the future.

**Reproducibility Statement.**    To facilitate the reproducibility of our algorithms and results, (i) we include a link to downloadable source code in supplementary materials, (ii) for our theoretical statements and results, we include clear explanations of any assumptions and a complete proof of the claims in Appendix D; for any datasets used in the experiments, a complete description of the data processing steps is in Appendix E. We will also release model checkpoints for all our models.

**Ethics Statement.**    Our work seeks to understand the fundamental capabilities and limitations of newly-emerging model architectures. As the amount of data and model size grows, we also week to understand how to make training these models more efficient—and run inference more efficiently. This potentially connects to energy savings during model development and deployment. We also note that

the relative underutilization of tensor cores in the FFT convolutions of state space models (even with our block FFT) suggests that consumer GPUs may be able to train models at a cheaper price point.

However, as with any language model training, developing new techniques may impact a wide range of applications, each with potential benefits and harms. For example, making language model training cheaper and making inference more efficient make it cheaper to spread disinformation. Similarly, improving the efficiency of model training may not reduce the overall environmental footprint of training, since the same resources may be used to train more models, or train the same models for longer. While our work makes partial progress on the fronts of efficiency and understanding, it does not explicitly address the issues of fairness and bias in language models.

## ACKNOWLEDGMENTS

We thank Albert Gu for helpful discussion regarding the model architecture, and more importantly for sending us daily hippo videos. We thank Together Computer for providing portions of the compute used to train models in this paper. We gratefully acknowledge the support of NIH under No. U54EB020405 (Mobilize), NSF under Nos. CCF1763315 (Beyond Sparsity), CCF1563078 (Volume to Velocity), and 1937301 (RTML); US DEVCOM ARL under No. W911NF-21-2-0251 (Interactive Human-AI Teaming); ONR under No. N000141712266 (Unifying Weak Supervision); ONR N00014-20-1-2480: Understanding and Applying Non-Euclidean Geometry in Machine Learning; N000142012275 (NEPTUNE); NXP, Xilinx, LETI-CEA, Intel, IBM, Microsoft, NEC, Toshiba, TSMC, ARM, Hitachi, BASF, Accenture, Ericsson, Qualcomm, Analog Devices, Google Cloud, Salesforce, Total, the HAI-GCP Cloud Credits for Research program, the Stanford Data Science Initiative (SDSI), Department of Defense (DoD) through the National Defense Science and Engineering Graduate Fellowship (NDSEG) Program, Wu Tsai Neuroscience Stanford Interdisciplinary Graduate Fellowship, and members of the Stanford DAWN project: Facebook, Google, and VMWare. The U.S. Government is authorized to reproduce and distribute reprints for Governmental purposes notwithstanding any copyright notation thereon. Any opinions, findings, and conclusions or recommendations expressed in this material are those of the authors and do not necessarily reflect the views, policies, or endorsements, either expressed or implied, of DARPA, NIH, ONR, or the U.S. Government. Atri Rudra's research is supported by NSF grant CCF-1763481.

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

# A    RELATED WORK

**State space models** have shown promise in modeling sequential data, including time series data (Gu et al., 2022a), audio (Goel et al., 2022), and visual data (Nguyen et al., 2022). Our model builds off work on simplifying and parameterizing diagonal versions of S4 (Gu et al., 2022b; Gupta et al., 2022; Gu et al., 2022c). Gated state spaces (Mehta et al., 2022) also aim to adapt SSMs to language modeling, but our results suggest that the GSS model does not perform as well as H3 (or even as well as earlier SSMs like S4D). The idea to combine SSMs with attention in hybrid models is not new; Mehta et al. (Mehta et al., 2022) also showed that interleaving attention with their GSS layer can improve performance, which we also validate on our OpenWebText experiments. These positive results suggest that attention and SSMs are complementary, and that hybrid models may be a promising direction for future work.

**Large language foundation models** (Bommasani et al., 2021) have demonstrated the power of scaling attention-based networks to billions of parameters and training them on trillions of tokens (Hoffmann et al., 2022). Understanding the mechanistic basis (Elhage et al., 2021) behind these models may yield insights into better design choices for future models. These and similar explorations have informed the design of H3 and our selection of synthetic languages. A number of recent works have also explored how to address the shortcomings of attention by approximating the attention computation (Wang et al., 2020; Katharopoulos et al., 2020; Choromanski et al., 2020; Tay et al., 2020; Kitaev et al., 2020; Daras et al., 2020). We believe these efforts are complementary to SSMs, and we are excited to see how they can be combined in future work.

**Linear attention** (Katharopoulos et al., 2020) and classical sequence models like RNNs serve as inspiration for H3. Appendix B draws a direct connection between linear attention and LTI systems. Luo et al. (Luo et al., 2021) also propose a variant of linear attention that can achieve $O(n \log n)$ scaling in sequence length. Appendix F evaluates linear attention on language modeling, and finds that it underperforms exact attention, whereas H3 outperforms attention. The multiplicative interactions in H3 are reminiscent of gating mechanisms in LSTMs (Hochreiter & Schmidhuber, 1996) and GRUs (Cho et al., 2014), which suggests that architectural lessons from these sequence models may be useful for adapting SSMs to language modeling. A number of algorithms for scaling attention to longer sequences have also been proposed, such as Transformer-XL (Dai et al., 2019), Reformer (Kitaev et al., 2020), Performer (Choromanski et al., 2020), and Perceiver AR (Hawthorne et al., 2022). Some of these approaches underperform exact attention on language modeling, and may be slower in wall-clock speed (Dao et al., 2022b). A thorough comparison of these alternatives to exact attention and how well they scale in model size and amount of training data is fruitful future work.

**FFT** algorithms are used in a wide variety of applications, including signal processing (Oppenheim, 1978), control theory (Brogan, 1974), and more. Various algorithms for computing the FFT have existed for decades (Oppenheim et al., 2001). We hope our work on appealing to these classic algorithms to accelerate new applications such as learned SSMs will inspire future algorithmic exploration, even if hardware is not designed for them (Hooker, 2021).

# B    LINEAR ATTENTION AND TIME-VARYING SYSTEMS

We draw some connections from linear attention to LTI systems and SSMs.

We first present linear attention as a linear time-varying system, and show how converting it to a linear time-invariant system matches H3.

**Linear time-varying system and linear attention**    In general, a layer in a sequence model takes in a sequence and outputs a sequence. Many of these take the form of a linear time-varying system (thanks to the Picard-Lindelof theorem, nonlinear systems can be approximated by a series of linear system):

$$x_i = \mathbf{A}_i x_{i-1} + \mathbf{B}_i u_i,$$
$$y_i = \mathbf{C}_i x_i + \mathbf{D}_i u_i.$$

This has the same form as SSMs (Section 2), except that the matrices can depend on the timestep.

Recall the form of linear attention from Section 2. For the purpose of approximation, we ignore the denominator in linear attention Section 2 (i.e., assuming $d_i = 1$). We see that $S_i$ is just a cumulative sum, satisfying the recurrence $S_{i+1} = S_i + \phi(K_{i+1})V_{i+1}^T$. Similarly, $O_i$ satisfies the recurrence $O_{i+1} = \phi(Q_{i+1})^T S_{i+1}$. This is a linear time-varying system of the form $x_{i+1} = \mathbf{A}x_i + \mathbf{B}u_{i+1}$ and $y_{i+1} = \mathbf{C}_{i+1}x_{i+1}$ (with $\mathbf{A} = I$, $\mathbf{B} = I$, $u_i = \phi(K_i)V_i^T$, $C_i = \phi(Q_i)^T$). That is, $\mathbf{A}$ and $\mathbf{B}$ are constant, but $C$ is time-variant.

To convert this into a linear time-invariant version, we treat the time-variant $\mathbf{C}_i$ as a post-processing step. We instead of a fixed $\mathbf{C}$ for the LTI. This yields an LTI:

$$x_{i+1} = \mathbf{A}x_i + \mathbf{B}\phi(K_i)V_i^T,$$
$$y_{i+1} = \mathbf{C}x_i,$$

for some matrices $\mathbf{A},\mathbf{B},\mathbf{C}$ that are learned. We then apply post-processing by multiply $y_{i+1}$ with $\phi(Q_i)^T$. Replacing $\phi(K_i)$ with a shift SSM yields an analogue to H3.

## C   METHOD DETAILS

Since we have described the forward pass in Section 3, we describe here the backward pass in details.

### C.1   BACKWARD PASS

We show how to compute the backward pass in a fused kernel.

Let $y = f * u + \mathbf{D}u$. In our case, we have $f$ and $u$ have the same length, so they are symmetric as far as the convolution is concerned.

Suppose we are given $dy = \frac{\partial l}{\partial y}$ (where $l$ is some loss function). We wish to compute $du$, $df$, and $dD$ (which are $\frac{\partial l}{\partial u}$, $\frac{\partial l}{\partial f}$, and $\frac{\partial l}{\partial \mathbf{D}}$, respectively).

The most challenging part is computing the gradient through the convolution operator - but we'll see that we can re-use our FFT infrastructure for it. The rest of the operations are straightforward; we have $d\mathbf{D} = dyu^T$.

**Gradient of the Convolution**   Here, we'll discuss how to compute $df$ by integrating w.r.t. the convolution operator $*$. As an immediate consequence, we'll be able to compute $du$ as well.

Since $f$ and $u$ are the same length $L$, $f * u$ and $u * f$ have the same result. Thus, we'll start from $u * f$ here.

For some notation, let $O = u * f$. Then, $dO = dy$. Recall that $O[i] = \sum_{j=0}^{i-1} u[i-j]f[j]$.

We'll start by extending $u$ and $f$ with zeros, to give them length $2L$. Let $u' = [u[0], u[1], ..., u[L-1], 0, ..., 0]$, and $f'$ extended similarly. Let $O' = u' * f'$, and $O = O'[:N]$. Assume that we have all the values of $dO'$ (we only have them for the first half, but we'll see that it doesn't matter in the end).

Let's construct a Toeplitz matrix $H_{u'}$ such that $u' * f' = H_{u'}f'$:

$$H_{u'} = \begin{bmatrix} u'[0] & 0 & ... & 0 \\ u'[1] & u'[0] & ... & 0 \\ \vdots & \vdots & \ddots & \vdots \\ u'[2L-1] & u'[2L-2] & ... & u'[0] \end{bmatrix}$$

Since, we have $u'[i] = f'[i] = 0$ for $i \geq L$, we can actually fill in the zeros of the above matrix as well:

$$H_{u'} = \begin{bmatrix} u'[0] & u'[2L-1] & ... & u'[1] \\ u'[1] & u'[0] & ... & u'[2] \\ \vdots & \vdots & \ddots & \vdots \\ u'[2L-1] & u'[2L-2] & ... & u'[0] \end{bmatrix}$$

Then, we can use the matrix multiplication chain rule to find that:

$$df' = H_{u'}^T dO' = \begin{bmatrix} u'[0] & u'[1] & ... & u'[2L-1] \\ u'[2L-1] & u'[0] & ... & u'[2L-2] \\ \vdots & \vdots & \ddots & \vdots \\ u'[1] & u'[2] & ... & u'[0] \end{bmatrix}$$

$$= \begin{bmatrix} u'[0] & u'[-(2L-1)] & ... & u'[-1] \\ u'[-1] & u'[0] & ... & u'[-2] \\ \vdots & \vdots & \ddots & \vdots \\ u'[-(2L-1)] & u'[-(2L-2)] & ... & u'[0] \end{bmatrix},$$

where we use $u'[-i]$ to mean $u'[2L - i]$. Notice that this matrix has the same format as $H_{u'}$! Let $u'_* = [u'[0], u'[-1], ..., u'[-(2N-1)]]$. Then:

$$df' = (u'_* * dO').$$

So how do we compute $u'_*$ efficiently? Naively, we might incur some nasty memory access issues. But a nice property about the DFT saves us!

Let $U[i]$ be the $i$-th element of the DFT of a signal $u$. Note that $U[i]$ is complex. We have:

$$U^*[i] = U[-i],$$

where here the $*$ represents the complex conjugate. We can use this property to compute $df'$ efficiently:
$$df' = u'_* * dO' = iFFT(FFT^*(u')FFT(dO')) \Rightarrow df = df'[:N] = iFFT(FFT^*(u')FFT(dy'))[:N],$$
where $FFT^*$ denotes taking the complex conjugate of the FFT, and $dy'$ denotes $dy$, padded with zeros.

**Computing $du$**   We can use this same trick to compute $du$, except we need to add in the contribution from the original $\mathbf{D}u$ term. We end up with:
$$du = du'[:N] + \mathbf{D}dy = iFFT(FFT^*(f')FFT(dy'))[:N] + \mathbf{D}dy.$$

### C.2   State-Passing Matrices

We show how to derive $\mathbf{M}_{ux}$ for the state update in our state-passing algorithm.

We wish to construct a matrix $vM_{ux} \in \mathbb{R}^{m \times N'}$ such that $\mathbf{M}_{ux}u = \sum_{i=1}^{N'} \mathbf{A}^{N'-1}\mathbf{B}u_i$. Note that $\mathbf{A}^i\mathbf{B} \in \mathbb{R}^{d \times 1}$ is a column vector. We can simply stack these column vectors to form a matrix: $\mathbf{M}_{ux} = [\mathbf{A}^{N'-1}\mathbf{B}, \mathbf{A}^{N'-2}\mathbf{B}, ..., \mathbf{B}]$.

## D   Proofs

We show parameterizations of H3 and attention that solves the associative recall task. We prove Proposition 1 and Proposition 2.

### D.1   H3 Expressivity

This section formally describes a parameterization of H3 that solves the associative recall task.

#### D.1.1   Example Language $\Lambda$

Consider a simple language with 4 keys and 4 values. For concreteness, we will use the keys $\{k_1, k_2, k_3, k_4\} = L_K$ and the values $\{v_1, v_2, v_3, v_4\} = L_V$, i.e. our language $L = L_K \cup L_V$. Given a sequence of key-value pairs with one key at the end, we want a model to generate the value associated with the key at the end. Assume that the key at the end appeared in the sequence.

More formally, let $N+1$ be the length of the sequence, $N$ even. The language $\Lambda$ consists of sequences $x \in L^{N+1}$. Each sequence has an associated mapping $f_x : L_K \to L_V$. For each sequence, the odd indices are randomly sampled from $L_K$, for $x_1, x_3, ..., x_{N-1}$. The even indices are defined by $f_x$: $x_{2*i} = f_x(x_{2*i-1})$, for $1 \le i \le N/2$. The last item in the sequence $x_{N+1}$, is randomly drawn from the keys that have appeared in $x$ already, i.e. $x_{N+1} \in \cup\{x_1, x_3, ..., x_{N-1}\}$. The goal of this language modeling task is to produce $f_x(x_{N+1})$ at the end of the sequence.

#### D.1.2   H3 Model to Solve $\Lambda$

We describe a toy H3 model that can solve $\Lambda$.

Consider a model consisting of an embedding layer, an H3 model, and an output projection with softmax. Recall that $d$ is the dimension of the H3 model, $m$ is the dimension of its hidden states, and $H$ is the number of heads. Let $d=8, m=2, H=4$. Let the embedding layer map each key $k_i$ to the $e_i$ basis vector, and map each value $v_i$ to the $e_{4+i}$ basis vector.

Let $\mathbf{B}_{shift}$ and $\mathbf{C}_{shift}$ denote the parameters of the shift SSM, and $\mathbf{A}_{diag}$, $\mathbf{B}_{diag}$, and $\mathbf{C}_{diag}$ denote the parameters of the diagonal SSM (let $\mathbf{D}$ be zero for both). Let $\mathbf{B}_{shift} = \mathbf{B}_{diag} = \mathbf{C}_{diag} = e_1$. Let $\mathbf{C}_{shift} = [01]$. Let $\mathbf{A}_{diag}$ be a diagonal matrix with 1s along its diagonal for each H3.

**Remark.** *The action of a diagonal SSM parameterized by $\mathbf{A}_{diag}$, $\mathbf{B}_{diag}$, and $\mathbf{C}_{diag}$ is to act as a cumulative sum over all its input. The action of shift SSM parameterized by $\mathbf{B}_{shift}$ and $\mathbf{C}_{shift}$ is to shift its input by one time step.*

Recall that the H3 layer maps its input to $Q$, $K$, and $V$ by applying $u\mathbf{W}_Q$, $u\mathbf{W}_K$, and $u\mathbf{W}_V$. Let $\mathbf{W}_Q$ and $\mathbf{W}_K$ be the following:

$$\mathbf{W}_Q = \mathbf{W}_K = \begin{bmatrix} 1 & 1 & 0 & 0 & 0 & 0 & 0 & 0 \\ 0 & 0 & 1 & 1 & 0 & 0 & 0 & 0 \\ 0 & 0 & 0 & 0 & 1 & 1 & 0 & 0 \\ 0 & 0 & 0 & 0 & 0 & 0 & 1 & 1 \\ 0 & 0 & 0 & 0 & 0 & 0 & 0 & 0 \\ 0 & 0 & 0 & 0 & 0 & 0 & 0 & 0 \\ 0 & 0 & 0 & 0 & 0 & 0 & 0 & 0 \\ 0 & 0 & 0 & 0 & 0 & 0 & 0 & 0 \end{bmatrix}$$

Recall that $Q$ and $K$ are split into $H$ heads ($\mathbf{Q}^{(i)}, \mathbf{K}^{(i)}$ for $i \in \{1,2,3,4\}$), each of which is sent to an independent H3.

**Remark.** *The action of $\mathbf{W}_Q$ and $\mathbf{W}_K$ are to "assign" each key to a different H3 head, i.e., $\mathbf{Q}_t^{(i)}$ is only non-zero when $x_t = k_i$. Similarly, $\overline{\mathbf{K}}_t^{(i)}$ is only non-zero when $x_{t-1} = k_i$ (since $\overline{\mathbf{K}}_t = \mathbf{K}_{t-1}$ due to the time delay of the shift SSM).*

Let $\mathbf{W}_V$ be the following:

$$\mathbf{W}_V = \begin{bmatrix} 0 & 0 & 0 & 0 & 0 & 0 & 0 & 0 \\ 0 & 0 & 0 & 0 & 0 & 0 & 0 & 0 \\ 0 & 0 & 0 & 0 & 0 & 0 & 0 & 0 \\ 0 & 0 & 0 & 0 & 0 & 0 & 0 & 0 \\ 0 & 0 & 0 & 0 & 0 & 0 & 0 & 0 \\ 0 & 1 & 0 & 1 & 0 & 1 & 0 & 1 \\ 1 & 0 & 1 & 0 & 1 & 0 & 1 & 0 \\ 1 & 1 & 1 & 1 & 1 & 1 & 1 & 1 \end{bmatrix}$$

**Remark.** *The action of this matrix is to encode the input value (as "binary"), and send it to all H3 heads. E.g., $\mathbf{V}_t^{(1)} = \mathbf{V}_t^{(2)} = \mathbf{V}_t^{(3)} = \mathbf{V}_t^{(4)}$ for all $i$, and $\mathbf{V}_t^{(i)} = [0,0] \Leftrightarrow x_t = v_1$, $\mathbf{V}_t^{(i)} = [0,1] \Leftrightarrow x_t = v_2$, etc.*

We claim that for $x_{N+1} = k_i$, $\mathbf{O}_{N+1}^{(i)}$ will be a multiple of the binary encoding of $f_x(k_i)$, and all the other heads of the output $\mathbf{O}_{N+1}^{(j)}, 1 \leq j \leq 4, j \neq i$, will be zero. Let the output projection $\mathbf{W}_O$ be such that, with a non-linearity afterwards, it inverts the binary encoding to produce the embedding of the desired output $f_x(k_i)$. We will assume such a projection exists, proof left to the reader.

**Proposition 3.** *The model described above solves the associative recall problem for the language $\Lambda$.*

*Proof.* Proof sketch. WLOG, let $x_{N+1} = k_i$. Then $\mathbf{Q}^{(i)} = [1,1]$, but $\mathbf{Q}^{(j)} = [0,0]$ for $j \neq i$. Thus, $\mathbf{O}^{(j)} = [0,0]$ for $j \neq i$ due to the multiplicative interaction.

Since $\mathbf{Q}^{(i)} = [1,1]$, $\mathbf{O}^{(i)}$ is the output of the diag SSMs in the H3 head corresponding to $k_i$ (recall that each head has two independent shift SSMs and two independent diag SSMs). The output of the diag SSMs are the cumulative sum of all the inputs they have seen in the sequence.

For one of the diag SSMs to see a non-zero input, its preceding shift SSM must have a non-zero output. The only times $t$ this can happen in the sequence are when $x_{t-1} = k_i$. But then $x_t = f_x(k_i)$. Thus, the input to the diag SSMs are precisely the binary encoding of $f_x(k_i)$. Then the output $\mathbf{O}^{(i)}$ is a multiple of the binary encoding of $f_x(k_i)$, $\mathbf{W}_O$ decodes this output to the embedding of $f_x(k_i)$. □

### D.2 ATTENTION EXPRESSIVITY

We provide an informal sketch of a two-layer attention model that can solve the associative recall task, inspired by the construction of (Olsson et al., 2022). The first layer of the attention model outputs the embedding of the previous token in the sequence, and concatenates it with the current token in the sequence. The second layer compares the current token to the previous token embeddings, and outputs the paired embedding when there is a match—which is exactly the key-value lookup.

The construction proceeds as follows:

- In the first layer, let $Q_i$ be mapped to the positional embedding of token $x_{i-1}$ (e.g., $p_{i-1}$ if $p_i$ denotes the positional embedding of token $x_i$), and $K_i$ be mapped to the positional embedding of token $x_i$.
- The attention matrix $A$ is computed as $QK^T$, with a causal mask (i.e., $A_{i,j} = 0$ if $j > i$).
- Then, $softmax(A)$ approximates the shift matrix (see Section 3).
- Let $V_i$ be an encoding of token $x_i$, constrained to the first half of the hidden dimension.
- Then, for output $O = softmax(QK^T)V$, the first half of the vector $O_i$ is the encoding of token $x_{i-1}$.
- In the second layer, assume that you have a skip connection, that maps the encoding of the input token $x_i$ to the second half of the vector $O_i$.
- Then, the input to the second layer encodes both $x_{i-1}$ and $x_i$.
- In the second layer, let $Q_i$ extract the encoding of $x_i$, and let $K_i$ extract the encoding of $x_{i-1}$.

- Apply a causal mask on $QK^T$. Then, the value of $softmax(QK^T)_{i,j}$ is large if $x_i = x_{j-1}$, and $i > j-1$.

- Let $V_i$ extract the encoding of $x_i$.

- Then, output $O_i$ is the sum of values $x_j$ such as $x_{j-1} = x_i$. But then $O_i$ is exactly a lookup of the token that came after $x_i$ when it appeared previously in the sequence—which exactly solves associative recall.

We note that the above construction requires the ability for the positional encodings to select the previous token based on the dot product and softmax, and for token comparisons through the dot product and softmax.

### D.3 H3 COMPLEXITY

We prove Proposition 1, which states that the H3 layer takes $O(d^2N + dN\log N)$ time and $O(dN)$ space for sequence length $N$ and hidden dimension $d$.

*Proof.* We first analyze the time complexity. Consider the matrix multiplies in H3, where the input $u \in \mathbb{R}^{N \times d}$ is multiplied by three weight matrices of size $d \times d$. These take time $O(d^2N)$. The output $\mathbf{O}$ is also multiplied with an output projection weight matrix of size $d \times d$, also taking time $O(d^2N)$. Therefore the matrix multiplies take time $O(d^2N)$.

Now consider the two SSMs in H3. The first SSM involves a convolution of $\mathbf{K} \in \mathbb{R}^{N \times d}$ (in the $N$-dimension) with a kernel of size $N \times d$. This reduces to an FFT, a pointwise multiply, and an inverse FFT (in the $N$-dimension). This takes time $O(dN\log N)$. The second SSM involves $H$ convolutions, inputs of size $N \times d_h \times d_h$, along the $N$-dimension. This takes time:

$$O(Hd_h^2 N\log N) = O(dd_h N\log N) = O(dN\log N),$$

where we use the fact that $d_h = d/H$ and that $d_h = O(1)$. Therefore the two SSMs take total time $O(dN\log N)$. As a result, the H3 layer takes time:

$$O(d^2N + dN\log N).$$

Now we analyze the space complexity. The matrix multiplies all take space $O(dN)$. The FFTs, pointwise multiplies, and inverse FFTs of the two SSMs takes $O(dN)$ space and $O(Hd_h^2 N) = O(dd_h N) = O(dN)$ space. Therefore the overall space complexity is $O(dN)$.

$\square$

### D.4 STATE PASSING CORRECTNESS

We prove Proposition 2. We assume that the BLOCKFFTCONV algorithm is correct, i.e., the output $y = \text{BLOCKFFTCONV}(f,u)$ is equal to the output of an SSM with convolution kernel $f$ and input $u$.

*Proof.* Proof by induction on $C$.

**Base case:** $C = 1$. WTS $y = [y^{(1)}]$, $\mathbf{M}_{xx}x_{N'}^{(0)} + \mathbf{M}_{ux}u^{(1)} = x_N$.

In this case, note that $N = N'$. Then $y^{(1)} = \mathbf{M}_{xy}x_{N'}^{(0)} + \text{BLOCKFFTCONV}(f,u_1) = \text{BLOCKFFTCONV}(f,u_1)$. But $u = u_1$, so $y = y^{(1)} = [y^{(1)}]$.

Additionally, by the recursive definition of a state space,

$$
\begin{aligned}
x_N &= \mathbf{A}^{N-1}x_0 + \sum_{i=1}^{N} \mathbf{A}^{N-i}\mathbf{B}u_i \\
&= \mathbf{A}^{N'-1}x_0 + \sum_{i=1}^{N'} \mathbf{A}^{N'-i}\mathbf{B}u_i^{(1)} \\
&= \mathbf{M}_{xy}x_{N'}^{(0)} + [\mathbf{A}^{N'-1}\mathbf{B}, \mathbf{A}^{N'-2}\mathbf{B}, ..., \mathbf{B}]u^{(1)}. \\
&= \mathbf{M}_{xy}x_{N'}^{(0)} + \mathbf{M}_{ux}u^{(1)}.
\end{aligned}
$$

**Inductive step:** $C > 1$. Assume that $[y^{(1)},...,y^{(C-1)}] = y[:N'(C-1)]$, and $x_{N'}^{(C-1)} = x_{(C-1)N'}$. WTS that $y^{(C)} = y[N'(C-1):N'C]$, and $\mathbf{M}_{xx}x_{N'}^{(C-1)} + \mathbf{M}_{ux}u^{(C)} = x_N$. Let $t$ denote $N'(C-1)$.

For $i > (C-1)N'$, we have:

$$y_i = \mathbf{C}\mathbf{A}^{i-t}\mathbf{B}x_t + (f * [u_t, u_{t+1},...,u_{t+N'-1}])_{i-t} + \mathbf{D}u_i$$
$$= \mathbf{C}\mathbf{A}^{i-t}\mathbf{B}x_t + (f * u^{(C)})_{i-t} + \mathbf{D}u_i$$
$$= \mathbf{C}\mathbf{A}^{i-t}\mathbf{B}x_t + \text{BLOCKFFTCONV}(f, u^{(C)})_{i-N'}$$
$$= (\mathbf{M}_{xy}x_t + \text{BLOCKFFTCONV}(f, u^{(C)}))_{i-N'}$$
$$= (\mathbf{M}_{xy}x_{N'}^{(C-1)} + \text{BLOCKFFTCONV}(f, u^{(C)}))_{i-N'}$$
$$= y_{i-N'}^{(C)}.$$

Thus, $y^{(C)} = y[N'(C-1):N'C]$.

Similarly,

$$x_N = \mathbf{A}^{N'-1}x_{(C-1)N'} + \sum_{i=1}^{N'}\mathbf{A}^{N'-i}\mathbf{B}u_{i+t}$$
$$= \mathbf{A}^{N'-1}x_{N'}^{(C-1)} + \sum_{i=1}^{N'}\mathbf{A}^{N'-i}\mathbf{B}u_i^{(C)}$$
$$= \mathbf{M}_{xx}x_{N'}^{(C-1)} + [\mathbf{A}^{N'-1}\mathbf{B}, \mathbf{A}^{N'-2}\mathbf{B},...,\mathbf{B}]u^{(C)}$$
$$= \mathbf{M}_{xx}x_{N'}^{(C-1)} + \mathbf{M}_{ux}u^{(C)}.$$

$\square$

# E EXPERIMENTAL DETAILS

## E.1 SYNTHETICS

Our synthetic tasks, inspired by (Olsson et al., 2022), are designed to mimic the in-context learning capability of large language models—the ability to learn from examples in the input sequence, and use information from the input to generate the right answer for the output. For example, the induction head task requires memorizing the token that appears after the special ⊢ token in the input sequence, and the associative recall task requires learning the mapping from keys to tokens from the input sequence.

We evaluate synthetics by training two-layer versions of our GPT models, with different modules replacing attention. We train models with inner dimension 32, and MLP dimension 128. For all the synthetics, we use a learning rate of 5e-4 and a weight decay of 0.1. We sample 5000 training examples and 500 test examples from the same distribution, and we train for 200 epochs. Again, we use embedding dropout of 0.1 and residual dropout of 0.0.

## E.2 MODEL ARCHITECTURE

For our 125M models, we use 12 layers, with hidden dimension 1024, and MLP dimension 4096. For our 355M models, we use 24 layers, with the same hidden dimension and MLP dimension. The 1.3B models have 24 layers, with hidden dimension 2048, and MLP dimension 8192. The 2.7B models have 32 layers, hidden dimension 2560, and MLP dimension 10240. The hybrid models have 12, 16, 16, and 20 heads for the 125M, 355M, 1.3B, and 2.7B models, respectively. The 125M hybrid model has an attention layers at layers 1 and 7, the 355M and 1.3B hybrid models have attention layers at layers 1 and 13, and the 2.7B hybrid model has attention layers at layers 10 and 21. For both our hybrid models and our H3 models, we use SSM state size 64. Our hybrid model uses head dimension 1 for H3, while our pure H3 model uses head dimension 8. We run models with mixed-precision training, with bf16 for the MLP's and attention. When training language models, we use fp32 for the FFTConv.

## E.3 OPENWEBTEXT TRAINING

For the 125M models trained on OpenWebText, we follow the training recipe of the Megatron-LM repo.

We use an effective batch size of 512, and use gradient accumulation to fit into available GPU memory. We use the AdamW optimizer, with learning rate 6e-4 for GPT-2 small and 1.5e-4 for GPT-2 medium, and weight decay of 0.1. All models are trained with the same hyperparameters for 100K steps. We run all implementations with mixed-precision training (PyTorch AMP). We train models with sequence length 1024.

We use the Openwebtext dataset, with the GPT-2 BPE tokenizer. We randomly select 0.5% of the dataset as the validation set, with the rest being used as training set. This random selection of validation set is done once, and all models are evaluated on the same validation set.

### E.4 THE PILE TRAINING

For the 125M and 355M models trained on the Pile, we follow the training recipe of GPT-3. We use batch size 256, with sequence length 2048. We train our models for 800K steps. We use residual dropout 0.0 and embedding dropout 0.1. We use the AdamW optimizer, with learning rate 6e-4 for the 125M model and 3e-4 for the 355M model, and a weight decay of 0.1. We use a cosine schedule with 8000 steps for linear warmup, and decay the learning rate to 10% by 300B tokens, then continue training at 10% learning rate for another 100B tokens. We suspect that there exist better hyperparameters for H3 language models, but we did not have the resources to tune them.

For the 1.3B models, we double the batch size to 512 (with sequence length 2048), again following the training recipe of GPT-3. The number of training steps are halved so that we train on the same number of tokens.

For the Pile dataset, we again use the GPT-2 BPE tokenizer, similar to GPT-3 and GPT-Neo.

### E.5 SUPERGLUE

We follow the prompts used in the GPT-3 paper (Brown et al., 2020). For rank classification on the binary classification tasks, we use yes/no for WSC, WIC, MultiRC, and BoolQ, and we use true/false for RTE. For CB, we use true/false/neither as the three choices. For COPA and ReCoRD, we use the continuations provided by the task.

### E.6 HARDWARE

All models were trained on either a single 16xA100-40GB node or a cluster of 8xA100-80GB nodes.

## F ADDITIONAL EXPERIMENTS

### F.1 LRA ACCURACY

We evaluate the accuracy of H3 on LRA. We compare accuracy to S4D (Gu et al., 2022b), since H3 uses an S4D kernel as a component in its layer. We use the same hyperparameters as S4D, and make the layer bidirectional by making two copies and running them in opposite directions.

Table 9: LRA performance of H3 compared to S4D.

| Model | ListOps | Text | Retrieval | Image | Pathfinder | Path-X | Avg |
|---|---|---|---|---|---|---|---|
| S4D (Gu et al., 2022b) | **58.3** | 87.3 | 90.7 | **87.5** | **93.6** | **92.3** | **85.0** |
| H3 | 57.5 | **88.2** | **91.0** | 87.3 | 93.0 | 91.8 | 84.8 |

Table 9 shows that H3 performs well on the LRA benchmark, even thought it was designed for autoregressive language modeling. H3 outperforms S4D on two of the LRA tasks, and comes within 1 point on the others.

### F.2 WIKITEXT103

We train 125M-sized models on WikiText103 (Merity et al., 2016) and compare their test PPL to transformers, as well as other variants of efficient or long-range attention. We use the same hyperparameters and setup as training on OpenWebText. We also provide results from Transformer-XL and Perceiver AR for context, though the results may not be directly comparable due to differences in model size and tokenizer.

Table 10: Test PPL on WikiText103.

| Models | PPL |
|---|---|
| Transformer (125M) | 18.6 |
| Hybrid H3 (125M) | 18.5 |
| Performer (125M) (Choromanski et al., 2020) | 26.8 |
| Reformer (125M) (Kitaev et al., 2020) | 26.0 |
| Linear Attention (125M) (Katharopoulos et al., 2020) | 25.6 |
| Perceiver AR (358M) (Hawthorne et al., 2022) | 18.4 |
| Transformer-XL (285M) (Dai et al., 2019) | 18.4 |

Table 10 shows that the Hybrid H3 model is competitive with Transformers of the same size, as well as larger models such as the 358M Perceiver AR and 285M Transformer-XL. The hybrid H3 model also significantly outperforms transformers with performer, reformer, and linear attention.

We note that the Transformer-XL and Perceiver AR PPl numbers are from the original papers, and may not be directly comparable to our results. In particular, they use a tokenizer with a different vocab size,

which means that the PPLs are not directly comparable. In addition, the larger vocab size necessitates a change in the model (adaptive softmax) that may affect performance. The top five numbers in Table 10 are trained with the same setup and are directly comparable to each other.

### F.3 PG-19

We evaluate models trained on the PG-19 dataset (Rae et al., 2019), a natural language dataset comprised of texts from books. We compare the performance of Hybrid H3 compared against Transformers and linear attention. We use the same setup as evaluating on OpenWebText.

Table 11: Test PPL on PG-19.

| Models | PPL |
|---|---|
| Transformer (125M) | 17.0 |
| Hybrid H3 (125M) | 16.2 |
| Linear Attention (125M) (Katharopoulos et al., 2020) | 19.1 |

Table 11 shows that Hybrid H3 outperforms transformers and linear attention.

### F.4 LENGTH EXTRAPOLATION

One property of SSMs is that they can naturally extrapolate to sequence lengths longer than those seen during training. We use the synthetic associative recall task to demonstrate that H3 maintains this capability. To do so, we train a two-layer H3 model on sequences of length 20 drawn from the associative recall synthetic language. Then, we evaluate accuracy of the last token prediction on sequences of length 20 and 40.

Table 12: Accuracy of an H3 model trained for associative recall on sequences of length 20, evaluated on sequences of length 20 and 40.

| Models | Acc, seqlen 20 | Acc, seqlen 40 |
|---|---|---|
| H3 | 99.8 | 98.4 |

Table 12 shows that H3 maintains accuracy on sequences of length 40, which is twice the length of the training sequences.

### F.5 SCALING IN NUMBER OF TOKENS

We evaluate how well a Hybrid H3 model scales with the number of tokens seen during training, compared to a Transformer. For these experiments, we train a 125M Hybrid H3 model and a 125M Transformer on the Pile for 5B, 10B, and 15B tokens. We use a learning rate of 6e-4, adjusting the warmup to be 1% of the total training time, and adjusting the decay rate to decay the learning rate to 6e-5 by the end of training.

Table 13: Test PPL on the Pile for models trained with fewer tokens.

| Train Tokens | Hybrid H3 (125M) | Transformer (125M) |
|---|---|---|
| 5B | 11.8 | 12.7 |
| 10B | 10.7 | 11.3 |
| 15B | 10.2 | 10.7 |

Table 13 shows the results. Both the Hybrid H3 model and Transformer model improve as the number of training tokens increases.

### F.6 H3 LANGUAGE MODEL

Table 14: Zero-shot performance on SuperGLUE with rank classification. Best results for each model size in bold.

| Model | WSC | WIC | RTE | CB | MultiRC | ReCoRD | BoolQ | COPA | Average |
|---|---|---|---|---|---|---|---|---|---|
| OPT-125M | 39.4 | 52.0 | 48.7 | 37.4 | 58.9 | 44.9 | 59.6 | 60.0 | 50.1 |
| GPT-Neo-125M | 36.5 | **53.6** | 53.1 | 41.1 | **59.9** | 39.6 | **62.2** | 60.0 | 50.8 |
| **H3-125M** | **61.5** | 50.0 | 53.1 | 41.1 | 4.6 | 15.8 | 46.4 | 51.0 | 40.4 |
| **Hybrid H3-125M** | 39.4 | 51.4 | **59.2** | **48.2** | 51.4 | **55.0** | 59.6 | **67.0** | **53.9** |

Table 15: 3-shot performance on SuperGLUE with rank classification. Best results for each size in bold, second best underline.

| Model | WSC | WIC | RTE | CB | MultiRC | ReCoRD | BoolQ | COPA | Average |
|---|---|---|---|---|---|---|---|---|---|
| OPT-125M | 36.5 | **50.2** | 47.3 | 44.6 | **57.9** | 44.9 | 41.9 | 60.0 | 47.9 |
| GPT-Neo-125M | 38.5 | 50.0 | 53.1 | 17.9 | 56.3 | 39.6 | 62.1 | 60.0 | 47.2 |
| **H3-125M** | **63.5** | 50.0 | 52.3 | 48.2 | 32.6 | 15.8 | 37.8 | 51.0 | 43.9 |
| **Hybrid H3-125M** | 43.3 | 49.1 | **58.1** | 51.8 | 48.9 | **55.0** | 56.1 | **67.0** | **53.7** |

We report the results of a pure H3 language model on NLP evaluations. We train a 125M model on the Pile for 400B tokens. Tables 14 and 15 show zero-shot and few-shot performance on SuperGLUE, respectively.

## F.7 GENERATION PERFORMANCE

Table 16: Zero-shot performance on SuperGLUE with generation. Best results for each size in bold, second best underline.

| Model | WSC | WIC | RTE | CB | MultiRC | ReCoRD | BoolQ | COPA | Average |
|---|---|---|---|---|---|---|---|---|---|
| OPT-125M | **36.5** | **48.4** | **49.8** | 8.9 | **39.1** | 44.9 | 45.9 | 60.0 | **41.7** |
| GPT-Neo-125M | 27.9 | 11.3 | 45.8 | 8.9 | 19.1 | 39.6 | **56.4** | 60.0 | 33.6 |
| **Hybrid H3-125M** | 0.0 | 0.0 | 47.3 | 8.9 | 4.4 | 55.0 | 47.6 | 67.0 | 28.8 |
| GPT-2 medium (355M) | 50.0 | 50.2 | 16.2 | 21.4 | 10.5 | 53.3 | 38.4 | 65.0 | 38.1 |
| OPT-350M | 41.3 | 34.8 | **49.5** | 16.1 | **23.6** | 51.4 | 39.7 | 60.0 | **39.6** |
| **Hybrid H3-355M** | 22.1 | 21.5 | 47.3 | 8.9 | 17.1 | **62.3** | 44.4 | **69.0** | 36.6 |

Table 17: 3-shot performance on SuperGLUE with generation. Best results for each size in bold, second best underline.

| Model | WSC | WIC | RTE | CB | MultiRC | ReCoRD | BoolQ | COPA | Average |
|---|---|---|---|---|---|---|---|---|---|
| OPT-125M | 36.5 | 49.1 | 47.3 | 33.9 | 35.5 | 44.8 | 38.5 | 60.0 | 43.2 |
| GPT-Neo-125M | 38.5 | **50.0** | 53.1 | **42.9** | 22.5 | 39.7 | **61.2** | **68.0** | 47.0 |
| H3-125M | 0.0 | 0.0 | 47.3 | 8.9 | 0.0 | 15.4 | 37.8 | 53.0 | 20.3 |
| **Hybrid H3-125M** | **43.3** | 49.1 | **58.1** | 41.1 | **40.3** | **55.2** | 49.5 | 67.0 | **50.5** |
| GPT-2 medium (355M) | 36.5 | **50.5** | 47.3 | 28.6 | 35.3 | 53.1 | 37.8 | 63.0 | 44.0 |
| OPT-350M | 37.5 | 50.0 | 46.2 | 41.1 | 40.6 | 51.3 | 39.4 | 59.0 | 45.6 |
| **Hybrid H3-355M** | **42.3** | 47.5 | **50.5** | 37.5 | **57.5** | **61.4** | 45.4 | **73.0** | **51.9** |

We report results on SuperGLUE for generation. Instead of taking rank classification, we instead let the model generate a response, and we search for the gold label (i.e., "yes" or "no" for the yes/no questions) in the output. Tables 16 and 17 report the results. The trends for few-shot learning match with the logit results, but the hybrid and H3 models perform very poorly in zero-shot performance on some tasks. In these cases, the models tend to generate long text responses that are not relevant to the answer. The few-shot learning examples help the models generate answers in a parsable format.

## F.8 NON-TEXT SEQUENCE MODELING

We show that H3 outperforms Transformers on two non-text sequence modeling tasks: raw speech classification and seizure classification over raw EEG signals. H3 sets state-of-the-art performance on seizure classification and matches S4 on speech classification—which suggests that H3, or one of its hybrids, may be a strong candidate for a multimodal foundation model. Appendix E gives experimental details, and Appendix F gives an additional experiment on brain fMRI data.

**Seizure Classification from EEG** Seizures, which are characterized by uncontrolled brain activity, are some of the most common neurological disorders (Fisher et al., 2014). Chronic seizures, or epilepsy, cause a range of psychiatric and psycho-social disorders and impact the lives of roughly one percent of the global population (Kerr, 2012). The first step to treating epilepsy is manual analysis of scalp EEG by board-certified neurologists. However, the vast amount of EEG data produced by each patient (which can be up to days of data) makes manual EEG analysis a costly and time-consuming process.

To mitigate the costs associated with EEG monitoring, recent deep learning techniques have began to show promise in flagging abnormal EEG segments for potential seizure events (Siddiqui et al., 2020). A challenge with classifying EEG data is the trade-off between increasing input sequence length, where more context has been shown to improve seizure classification performance Saab et al. (2020), with the increased difficulty of training deep learning models on long sequences (e.g., an EEG signal sampled at 200Hz produces 12,000 time steps per minute). As a result, many techniques involve domain-specialized models and pre-processing steps, such as FFT transforms and graphical representations Tang et al. (2021).

We use the largest publicly available EEG corpus, TUSZ v1.5.2 (Shah et al., 2018), which includes 5,612 EEG signals from 636 patients, with 3,050 annotated seizures. Signals are segmented into 60-second clips, and split into train/val/test by patient. The train set contains 39765 clips, the val set contains 4351 clips, and the test set contains 10001 clips.

Table 18: Performance (AUROC) on 60s seizure classification from raw EEG (sequence length 12000).

| H3 | Transformer | Dense-CNN | CNN-LSTM | LSTM | 1D-CNN |
|---|---|---|---|---|---|
| **83.2** | x | 78.0 | 68.6 | 69.3 | 69.7 |

We evaluate binary seizure classification of 60-sec EEG clips, sampled at 200Hz, with 19 electrodes: $x \in R^{12,000 \times 19}$ and $y \in \{0,1\}$ on the TUSZ v1.5.2 (Shah et al., 2018) corpus. Transformers cannot process the long sequence length of EEG signals without running out of GPU memory, whereas H3 can—and sets state-of-the-art performance.

**Raw Speech Classification** The SC10 speech commands task (Warden, 2018) contains raw audio signals one second in length, sampled at 16kHz. Similarly to EEG signals, Transformers cannot process the long sequence length. Table 19 shows that H3 comes within half a point of S4, the state-of-the-art method.

Table 19: SC 10-class classification on raw audio (sequence length 16000).

| H3 | S4 | WaveGan-D | Transformer | Performer | CKConv |
|---|---|---|---|---|---|
| 97.04 | **97.50** | 96.25 | x | 30.77 | 71.66 |

**Functional Magnetic Resonance Imaging Data**   Functional Magnetic Resonance Imaging (fMRI) data are typically represented in four dimensions, indicating the measured blood-oxygen-level-dependent (BOLD) signal in temporal sequences $S = \{V_1, ..., V_t\}$ of 3-dimensional volumes $V \in \mathbb{R}^{x \times y \times z}$, each indicating the BOLD signal for all spatial locations of the brain (as defined by three spatial dimensions $x$, $y$, and $z$). A key challenge for the analysis of fMRI data is the high dimensionality and low sample size of its datasets, which typically contain many hundred thousand dimensions (i.e., voxels) for each of several hundred volumes $V$ in each of tens to hundreds of sequences $S$. In this setting, where the number of features exceed the number of samples, standard machine learning approaches are prone to overfitting.

In spite of the low sample size of individual datasets, neuroimaging research can be considered as recently entering a big data era because researchers more frequently share their collected datasets publicly (Markiewicz et al., 2021). The availability of these data open up the opportunity for pre-training in neuroimaging at scale, as recently demonstrated by (Thomas et al., 2022), enabling models to utilize the knowledge that can be learned from public functional neuroimaging data for the analysis of individual datasets. Specifically, (Thomas et al., 2022) evaluate the performance of several self-supervised learning frameworks for functional neuroimaging data by first pre-training models on a broad fMRI dataset spanning 11,980 fMRI runs from 1,726 individuals across 34 datasets and subsequently adapting the pre-trained models to two downstream mental state decoding datasets (namely, the HCP (Van Essen et al., 2013) and MDTB (King et al., 2019) datasets). In mental state decoding, predictive models are tasked with identifying (i.e., decoding) some set of mental states (e.g., answering questions about a prose story or math problem) from measured brain activity. The authors find that a GPT-based model, pre-trained in a causal learning framework, performs best in decoding the 20 (HCP) and 26 (MDTB) mental states of the two downstream datasets.

To evaluate the performance of H3 on fMRI data, we replicate this analysis, using the up- and downstream fMRI datasets that were published by (Thomas et al., 2022), treating H3 as a drop-in replacement for the GPT model. To alleviate the high dimensionality challenge of fMRI data, and due to the generally high spatial correlation of brain activity, the original authors have reduced the volumetric time series $S$ to a set $\Theta \in \theta_1, ..., \theta_n$ of $n = 1,024$ functionally-independent brain networks $\theta$ (as defined by the dictionaries of functional modes (DiFuMo) brain atlas (Dadi et al., 2020)), each describing the BOLD signal for some subset of voxels $v_{x,y,z} \in V$, such that the resulting sequences $X \in \mathbb{R}^{t \times n}$ describe the activity pattern of each brain network $n$ for time points $t$.

In line with (Thomas et al., 2022), we first pre-train models $f(\cdot)$ to predict the distribution of brain activity for the next time point $j$ of an input sequence $X$, using a mean absolute error ($L_{rec}$) training objective given the model's output $\hat{X} \in \mathbb{R}^{t \times n}$: $L_{rec} = \frac{1}{n} \sum_{i=1}^{n} |X_{j,i} - \hat{X}_{j,i}|$; $\hat{X}_{t,n} = b_n + \sum_n f(E^X)_{t,e} w_{e,n}$; $E_{t,e}^X = E^{TR} + E^{pos} + b_e + \sum_n X_{t,n} w_{n,e}$. Here, $E^{TR} \in \mathbb{R}^e$ and $E^{pos} \in \mathbb{R}^e$ represent learnable embeddings for each possible time point and position of an input sequence (for details, see (Thomas et al., 2022)). As the sampling frequency of fMRI is variable, the same position of an input sequence can correspond to different time points. Note that $f(\cdot)$ processes the input in a lower-dimensional embedding representation $E^X \in \mathbb{R}^{t \times e}$ (with $e = 768$ dimensions).

We evaluate two model architectures for $f(\cdot)$, namely, the GPT variant used in (Thomas et al., 2022), with 4 hidden layers and 12 attention heads, and a corresponding H3 variant with 4 hidden layers (with $H = 64$ and $m = 1$; see section 3). For both models, the sequence of hidden-states outputs of the last model layer are used to determine $\hat{X}$.

Just as (Thomas et al., 2022), we randomly divide the upstream data into distinct training and validation datasets by randomly designating 5% of the fMRI runs of each of the 34 upstream datasets as validation data (at a minimum of 2 runs per dataset) and using the rest of the runs for training. During upstream learning, we then randomly sample sequences of 10 to 100 time points from the fMRI runs and train models with the ADAM optimizer (with $\beta_1 = 0.9$, $\beta_2 = 0.999$, and $\epsilon = 1e^{-8}$) for 5,000 steps at a mini-batch size of 512, and a learning rate of $5e^{-4}$. We apply a linear learning rate decay schedule (with a warm-up phase of 1% of the total number of training steps), gradient norm clipping at 1.0, and $L2$-regularisation (weighted by 0.1). We also apply dropout at a rate of 0.1 for the GPT-based model (based on (Thomas et al., 2022)) and evaluate three dropout rates for H3: 0.1, 0.2, and 0.3.

We find that the H3 variant trained with 0.2 dropout performs on par with the GPT model, in terms of mean absolute error (Fig. 3), and therefore continue all further analyses with this model variant. We also find that

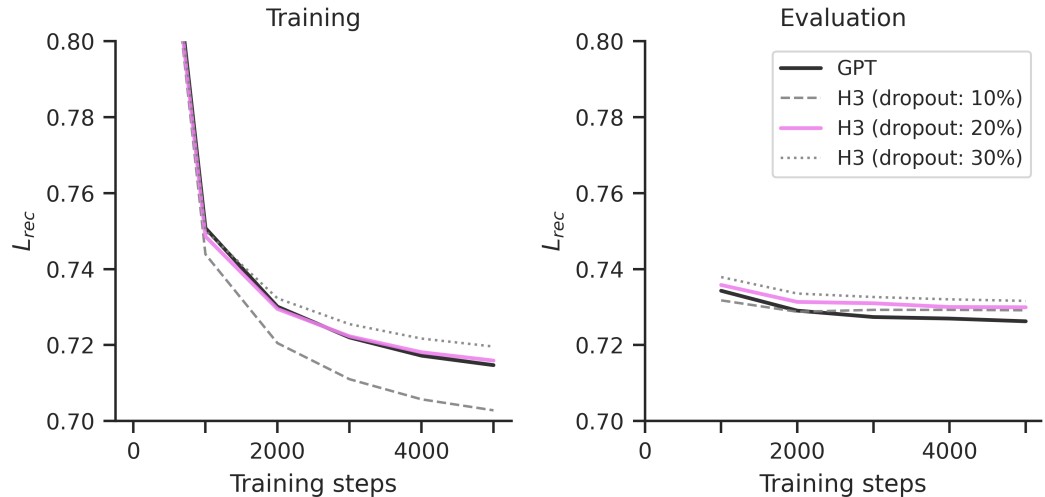

Figure 3: Upstream mean absolute error ($L_{rec}$) in training and evaluation datasets over the course of model training.

both models exhibit almost identify $L_{rec}$ error distributions throughout the brain, with relatively higher errors in the posterior parietal, occipital, and cingulate cortices as well parts of the limbic system (Fig. 4).

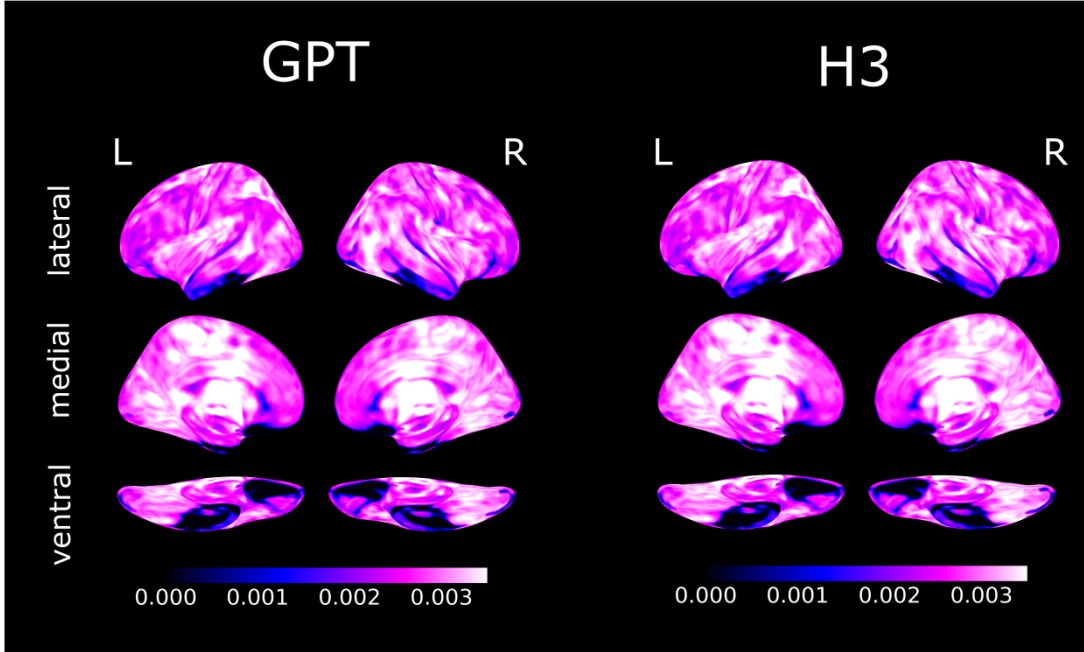

Figure 4: Mean absolute error ($L_{rec}$) of the final pre-trained models for each voxel of the brain projected onto the inflated cortical surface of the FsAverage template (Fischl, 2012).

To adapt the pre-trained models to mental state decoding, we add a learnable classification embedding $E^{cls} \in \mathbb{R}^n$ to the end of input sequences $X$ and forward the model's prediction $f(E^X)$ to a decoding head $p(\cdot)$, composed of a dense hidden layer with $e$ model units (one for each embedding dimension, with $tanh$ activation) as well as a $softmax$ output layer with one model unit $i$ for each considered mental state in the data. Accordingly, we adapt models by optimizing a standard cross entropy loss objective: $L_{cls} = -\sum_i y_i \log p(f(E^X))_i$, where $y_i$ indicates a binary variable that is 1 if $i$ is the correct mental state and 0 otherwise.

During downstream adaptation, we begin training with the respective pre-trained model parameters and then allow all parameters to change freely. Similar to (Thomas et al., 2022), we randomly split each of the two downstream datasets into distinct training and test datasets, each comprising 40 (HCP) or 10 (MDTB) distinct individuals. We adapt models for 750 steps at a mini-batch size of 256 and a learning rate of $5e^{-5}$

(otherwise using the same learning parameters as for upstream training). Importantly, we repeat each downstream training run 20 times using different random seeds, leading to different random splits of the data and variability in other non-deterministic training factors (such as random initialization and data shuffling).

As for the upstream data, the H3 and GPT-based models generally perform on par in their mental state decoding performances in the two left-out test datasets (Table 20).

Table 20: Downstream adaptation performance of models pre-trained on fMRI data, averaged over 20 training runs with varying random seeds. F1-scores are macro-averaged.

| Dataset | Model | Acc. ($\pm95\%$CI) | F1 ($\pm95\%$CI) |
|---------|-------|--------------------|------------------|
| HCP | GPT | 88.44 ($\pm0.39$) | 87.24 ($\pm0.39$) |
| | H3 | 88.75 ($\pm0.33$) | 87.16 ($\pm0.37$) |
| MDTB | GPT | 89.47 ($\pm0.44$) | 88.74 ($\pm0.54$) |
| | H3 | 88.25 ($\pm0.45$) | 85.76 ($\pm0.61$) |

