# OpenReview forum: "Hungry Hungry Hippos: Towards Language Modeling with State Space Models"
_ICLR.cc/2023/Conference — ICLR 2023 notable top 25%_

### Official Review · Reviewer_EkLp · 2022-10-25

**Confidence:** 4
**Clarity, Quality, Novelty And Reproducibility:** The paper is clear and the empirical …
**Correctness:** 3
**Technical Novelty And Significance:** 3
**Empirical Novelty And Significance:** 3
**Recommendation:** 6

**Strength And Weaknesses:**

Pros:

1. It is interesting to use synthetic tasks to investigate the drawback of the state-space model.
2. It is important to improve S4 on large-scale language modelling tasks. This work make an essential step towards this goal
3. The empirical results is adquante to support the advantage of the model

Cros:

Major questions:

1.The authors proposed two synthetic tasks, Induction Head and Associative Recall. However, the relationship between these two tasks and the language model performance remains unknown. Why does the poor performance of the language model have a strong connection with these two tasks? This connection should be discussed clearly since it motivates the modification, which (as the authors stated) leads to better language model performance.

2.The change improves the language modelling performance, but will the new model hurt the LRA performance. If I didn't miss anything, there is only an efficiency study on LRA without performance comparison. This work will be problematic if the model fails to work on LRA but only work on LMs.

Minors:

1. The authors claim that standard attention has the ability to complete Induction Head and Associative Recall. Is this a rigorous statement or not? If yes, I recommend the authors give formal proof of the statement. I am not that sure whether attention can succeed #for any length#

2. The authors were motivated by linear attention and proposed an O(nlogn) architecture with FFT. The work "Stable, Fast and Accurate: Kernelized Attention with Relative Positional Encoding" achieves the same efficiency with FFT based on linear attention. I believe the relationship should be discussed as a reference

**Summary Of The Paper:**

This work aims to improve the state space models (SSMs), which achieve state-of-the-art performance on the LRA benchmark but perform worse on language modeling and large-scale language pre-training. The authors identified two problems related to the model design through two synthetic tasks. Based on the observation, the authors further proposed some modifications which lead to strong LM performance. At the same time, the authors also leverage Flash to improve the efficiency of the inference.

**Summary Of The Review:**

The authors solved an important problem for state-space models, and the empirical study is solid. I hope the authors can address my concerns, and currently, I give a weak acceptance.

---

> ### Author Response · Authors · 2022-11-17
> **Response to Reviewer EkLp**
>
> We thank you for your feedback and comments. Answers to your specific questions below.
>
> **Major Question 1:**
>
> These two synthetic tasks were designed to evaluate the ability of a model to perform “in-context learning” – producing an output based on a particular computation over the inputs. We draw inspiration from work such as Olsson et al [1], as well as empirical observations about the abilities of large language models to have strong few-shot performance solely from examples in their input. As intuition – if a model cannot produce an output based on a computation over its input, it is hard to imagine that it can perform few-shot learning over its input.
>
> We have added additional discussion about the motivation behind these synthetic tasks in Appendix E.1.
>
> **Major Question 2:**
>
> Please see the common response for the LRA study. Even though H3 was designed with autoregressive language modeling in mind, it still performs well on the LRA benchmark, almost as well as recent S4 variants.
>
> **Minor Question 1:**
>
> Thank you for this suggestion. We have added a sketch of a two-layer Transformer in Appendix D.2 to show that it can solve associative recall, as long as the embedding dimension is large enough to encode tokens and the positional embedding.
>
> Roughly, the construction goes as follows:
> * In the first layer, $Q_i$ is mapped to the positional embedding of the previous token ($p_{i-1}$), and $K_i$ is mapped to the positional embedding of the current token ($p_i$). Apply a causal mask on $QK^T$.
> * Then, $softmax(QK^T)$ approximates a shift matrix.
> * Let $V_i$ be a direct encoding of the token $x_i$.
> * Then, for output $O = softmax(QK^T)V$, $O_i$ is the encoding of token $x_{i-1}$.
> * In the second layer, assume that you have a skip connection, such that the input encodes both $x_i$ and $x_{i-1}$.
> * Let $Q$ extract the encoding of $x_i$, and $K$ extract the encoding of $x_{i-1}$. Apply a causal mask on $QK^T$.
> * Then, for $A = softmax(QK^T)$, $A_{i,j}$ is 1 if $x_i = x_{j-1}$, and $i > j - 1$.
> * Finally, let $V_i$ contain the encodings of $x_i$.
> * Then, for output $O = AV$, $O_i$ is the sum of values $x_j$ such that $x_{j-1} = x_i$. But then $O_i$ is exactly the encoding of the answer.
>
> **Minor Question 2:**
>
> Thank you for pointing out this reference. We have added this reference to our related work, and added comparisons against linear/kernelized attention on WikiText103 and PG19 (please see the common response). Linear attention underperforms exact attention on language modeling.
>
> **References**
>
> [1] Olsson et al. In-Context Learning and Induction Heads. Transformer Circuits Thread, 2022.

---

### Official Review · Reviewer_PnYM · 2022-10-25

**Confidence:** 3
**Correctness:** 3
**Technical Novelty And Significance:** 4
**Empirical Novelty And Significance:** 3
**Recommendation:** 8

**Clarity, Quality, Novelty And Reproducibility:**

The paper is written clearly and H3 is a simple and novel extension of previous SSMs. While the implementation is not straightforward to derive from only the paper, the code is also open sourced.

**Strength And Weaknesses:**

**Strengths** I found the paper interesting and useful for thinking beyond typical transformer architectures. The paper proposes a simple SSM layer that can replace any self-attention module and broadly applicable. Experimental results are also promising and efficiency improvements should help other work in state space models.

**Weaknesses**
1. You mention that SSMs are capable of extrapolating to sequences longer than observed during training. I found no empirical results to support this claim in the paper.

2. There are missing references in long-term dependency line of work, especially *Perceiver* style models. Given that one of the things that the paper claims is capturing long term dependencies, I think these need to be cited and compared against.

3. How would you compare to other models in Table 2 is H3 is trained from data rather than by construction? Please also clarify how other models, excluding attention, are obtained.

4. One thing that Transformers have shown is they scale with data and model sizes. While from 125M to 335M, there is a clear improvement in Table 4, I would like to see if there is a similar pattern for data size as in Table 6.

General-purpose, long-context autoregressive modeling with Perceiver AR. Curtis Hawthorne, Andrew Jaegle, Cătălina Cangea, Sebastian Borgeaud, Charlie Nash, Mateusz Malinowski, Sander Dieleman, Oriol Vinyals, Matthew Botvinick, Ian Simon, Hannah Sheahan, Neil Zeghidour, Jean-Baptiste Alayrac, João Carreira, Jesse Engel.

**Summary Of The Paper:**

This paper presents a new state space model (SSM) that achieves on par with transformer based language models while inferring up to 2 times faster. The authors first highlight the drawback of previous SSMs compared to attention -- SSMs can't recall earlier tokens in the sequence and can't compare tokens across the sequence. The authors propose an improvement that captures these features by using `shift` and `diagonal` SSM layers. Shift layer shifts dimensions in an input (or steps if the input is a sequence) and diagonal SSM is convenient for computing kernel in the convolution. By borrowing the structure of linear attention, the authors propose a new SSM layer, called H3, that utilizes recurrence structure and can scale to longer sequences better. They show that H3 improves previous SSMs and achieves on par with transformers on synthetic as well as real OpenWebText tasks. They further propose a mechanism, called FastFFTConv, for efficient estimation of convolution where the convolution is replaced with `iFFT (FFT(u) * FFT(f))`. They divide input into chunks -- each chunk fits into the GPU memory -- and keep end-state of each chunk for a more efficient convolution which results in up to 3x speed up against attention.

**Summary Of The Review:**

H3 is a simple yet novel extension of previous SSMs. It presents improvements compared to Transformers and can infer 2 times faster. FastFFTConv is useful for efficient training and can be applied to other SSM works.

---

> ### Author Response · Authors · 2022-11-17
> **Response to Reviewer PnYM**
>
> We thank you for your positive comments and thoughtful questions. We answer your specific questions below.
>
> **Weakness 1:**
>
> Please see the common response for length extrapolation experiments.
>
> **Weakness 2:**
>
> Thank you for pointing out Perceiver AR. We have added a citation to our discussion of related work and added a comparison in our WikiText103 evaluation (please see common response).
>
> **Weakness 3:**
>
> In Table 2, all models are trained (not built by construction). For all models, we train using a next-token prediction loss on training data drawn from the synthetic language.
>
> **Weakness 4:**
>
> Please see the data size experiment in the common response.

---

### Official Review · Reviewer_UjQC · 2022-10-26

**Confidence:** 4
**Correctness:** 3
**Technical Novelty And Significance:** 3
**Empirical Novelty And Significance:** 2
**Recommendation:** 6

**Clarity, Quality, Novelty And Reproducibility:**

The experimental setup in this paper make it difficult to fairly compare H3 with previous models:

1. The authors trained H3 on the PILE data and calculated PPL on OpenWebText and WikiText103. But both the two datasets have their own training data, and several previous models have reported PPL on them. Why not following previous setup and directly comparing H3 with previous models, such as standard Transformer, Transformer-XL and other efficient Transformer architectures. It is strongly recommended to conducted language modeling experiments on standard benchmarks, such as WikiText103 and PG19, and closely follows previous settings for fair comparison.

2. For the comparison in Table4, since GPT2 models are not trained on the same PILE dataset, the PPLs are not directly comparable.

3. A lot model details are missed, making it hard to check or reproduce experimental results.

Other questions about experiments:
1. Does FlashFFTConv support half precision training?

2. Have you evaluated H3 accuracy on LRA?

**Strength And Weaknesses:**

Strengths: The design of H3 is novel and well-motivated and the paper writing is clear and easy to follow. The authors also clearly claimed their contributions.

Weaknesses: The main concerns are from experiments. I found that the experimental setup on language models is not standard, which makes the comparisons unfair. It is difficult to position H3 among other neural models in language modeling. I elaborated my concerns in the following questions.

**Summary Of The Paper:**

This paper proposed the H3 architecture to narrow the gap between state space models and attention models on language modeling. Basically, H3 uses a shift SSM and a diagonal SSM to replace the non-linear function in linear attention. The shift SSM is designed to remember tokens in the input sequence while the multiplication between the two SSM outputs for token comparison.
To further improve H3 efficiency on hardware the authors proposed FlashFFTConv, which uses fused block FFT algorithm.

Experiments were conducted on training multiple variants of H3 on the PILE dataset, and compared with corresponding GPT2 checkpoints, on OpenWebText and WikiText103. Moreover, the authors also evaluate H3 on zero-shot and few-shot classification.
At last, they evaluate efficiency of the proposed FlashFFTConv algorithm on the LRA benchmark, achieving about 2 times speed up.

**Summary Of The Review:**

To sum up, I am worried that the paper needs to re-organize its experiments and analysis before publication.

---

> ### Author Response · Authors · 2022-11-17
> **Response to Reviewer UjQC**
>
> Thank you for your detailed and thoughtful comments. They have helped to improve our paper. We answer some specific questions below.
>
> > But both the two datasets have their own training data, and several previous models have reported PPL on them.
>
> We report validation PPL from training on OpenWebText in Table 3. We have updated the paper to make this comparison more clear. We report PPL from training and evaluating directly on WikiText103 in the common response.
>
> We have also added comparisons against other efficient Transformer architectures and Transformer-XL in the common response.
>
> > For the comparison in Table4, since GPT2 models are not trained on the same PILE dataset, the PPLs are not directly comparable.
>
> Thank you for pointing this out. We have added a note to our paper to make this difference more clear.
>
> > A lot model details are missed, making it hard to check or reproduce experimental results.
>
> We have submitted code in the supplemental material, and described the model architecture in Appendix E.1. Can you specify which model details in particular would be helpful for reproduction?
>
> > Does FlashFFTConv support half precision training?
>
> Yes.
>
> > Have you evaluated H3 accuracy on LRA?
>
> Yes, please see the common response.

---

### Author Response · Authors · 2022-11-17
**General Response (1/2)**

We thank the reviewers for carefully reading our paper and providing thoughtful comments, which have improved our paper. We appreciate that the reviewers thought that our paper was “interesting” (reviewers PnYM and EkLp), “well-motivated” (reviewer UjQC), “important” (reviewer EkLp), with strong and promising experiments (reviewers UjQC, EkLp).

Here, we first briefly discuss our motivation for our evaluation strategy, and then present new experiments suggested by the reviewers on LRA accuracy (UjQC, EkLp), comparisons against other efficient attention architectures (UjQC, PnYM, EkLp), PG-19 PPL (UjQC), length extrapolation (PnYM), and data scaling (PnYM) experiments. We have added all these experiments to our paper. Overall, we find that H3 matches or outperforms Transformers, naturally extrapolates to sequences longer than seen during training, and scales with the amount of training data.

**Motivation for Training on the PILE**
Our goal for this paper was to train a language model that can compete with the best open-source language models available (within our compute capabilities). To that end, we tried to match as closely as possible the training setup of previous large language models, such as GPT-Neo, GPT-3, and GPT-2. Of these, GPT-Neo is the only effort that trained on entirely open-source data (the PILE), so we follow their setup for training data. We also follow the evaluation procedures of these efforts through few- and zero-shot performance on standard NLP benchmarks (SuperGLUE), and zero-shot PPL evaluation.

We are excited that we can outperform open-source Transformer-based language models of the same model sizes in this setup (Tables 4, 5, and 6 in our original submission).

For a more straight-forward comparison between H3 and Transformers (trained and evaluated the same datasets), we included results on OpenWebtext in Table 3 in our original submission, and we added additional comparisons as requested by the reviewers below.

**Long-Range Arena Accuracy (ujQC, EkLP)**
We report the accuracy of H3 on LRA, compared to S4D. Even though H3 was designed with autoregressive language modeling in mind, it still performs well on the LRA benchmark, almost as well as S4D [1], a recent S4 variant that we use for the second SSM in our H3 layer. We reproduced the setup from the S4D paper.

|   | **ListOps** | **Text** | **Retrieval** | **Image** | **Pathfinder** | **PathX** | **Avg** |
| ---: | :---: | :---: | :---: | :---: | :---: | :---: | :---: |
| S4D | 58.3 | 87.3 | 90.7 | 87.5 | 93.6 | 92.3 | 85.0 |
| H3 | 57.5 | 88.2 | 91.0 | 87.3 | 93.0 | 91.8 | 84.8 |

**Other Efficient Attention, WikiText103 (UjQC, PnYM, EkLp)**
We compare H3 against Transformer and other efficient attention architectures (Performer [2], Reformer [3], Linear Attention [4]) for training and evaluating on WikiText103. We report Perceiver AR [5] and Transformer-XL [6] for context, but the results may not be directly comparable due to differences in model size. We report test PPL in this table. Our Hybrid H3 model matches the performance of Transformers of the same size, and is better than the efficient attention baselines.

| **Model** | **WikiText103 PPL** |
| ---: | :---: |
| Transformer (125M) | 18.6 |
| Hybrid H3 (125M) | 18.5 |
| Performer (125M) | 26.8 |
| Reformer (125M) | 26.0 |
| Linear Attention (125M) | 25.6 |
| Perceiver AR* (358M) | 18.4 |
| Transformer-XL* (285M) | 18.4 |

**PG19 (UjQC)**
We report the test PPL for training and evaluating on the PG-19 dataset. We compare only against Transformer and Linear Attention due to the cost of training other baselines (both Performer and Reformer are significantly slower than vanilla attention). Our Hybrid H3 model outperforms both Transformers and Linear Attention.

| **Model** | **PG19 PPL** |
| ---: | :---: |
| Transformer (125M) | 17.0 |
| Hybrid H3 (125M) | 16.2 |
| Linear Attention (125M) | 19.1 |

**Length Extrapolation (PnYM)**
We show that H3 can extrapolate to longer sequences than seen in training using the associative recall synthetic language: we train on sequences of length 20, and evaluate on sequences of length 40. We generate sequences the same way as the synthetics in our original paper, and we provide a full explanation in Appendix F.4 of our updated paper.

| **Model** | **Acc, Seqlen 20** | **Acc, Seqlen 40** |
| ---: | :---: | :---: |
| H3 | 99.8 | 98.4 |

This proof-of-concept experiment shows that H3 extrapolates to sequence lengths longer than seen during training.

---

> ### Author Response · Authors · 2022-11-17
> **General Response (2/2)**
>
> **Data Scaling (PnYM)**
> We report the validation PPL of a 125M H3 model, trained with fewer tokens, compared to Transformers. We adjust the learning rate schedule accordingly to warm up in 1% of training time, and decay to 10% of the full learning rate by the end of training (full details in Appendix F.5 of our updated paper).
>
> | **# Tokens** | **Hybrid H3 PPL** | **Transformer PPL** |
> | ---: | :---: | :---: |
> | 5B | 12.3 | 13.3 |
> | 10B | 11.2 | 11.9 |
> | 15B | 10.7 | 11.2 |
>
> H3 scales with the amount of training data, similarly to Transformers.
>
> **References**
>
> [1] Gu et al. On the Parameterization and Initialization of Diagonal State Space Models. NeurIPS 2022.
>
> [2] Choromanski et al. Rethinking Attention with Performers. ICLR 2020.
>
> [3] Kitaev et al. Reformer: The Efficient Transformer. ICML 2020.
>
> [4] Katharopoulos et al. Transformers are RNNs: Fast Autoregressive Transformers with Linear Attention. ICML 2020.
>
> [5] Hawthorne et al. General-Purpose, Long-Context Autoregressive Modeling with Perceiver AR. ICML 2022.
>
> [6] Dai et al. Transformer-XL: Attentive Language Models Beyond a Fixed-Length Context. ACL 2019.

---

> ### Comment · Reviewer_UjQC · 2022-11-18
> **Questions about experiments on WikiText-103**
>
> Thanks for your great efforts on these additional results.
>
> I have some questions about the reported PPLs on WikiText-103.
>
> **WikiText-103**. The reported PPLs are a bit too good considering the model sizes (125M). According to the transformer-xl work, the standard Transformer model with 151M parameters obtained around **24**. In your table, a standard Transformer model with 125M parameters obtains **18.6** PPL, which is even comparable with Transformer-XL with 285M parameters. Could you explain why your PPLs are significantly better than previous results?

---

> > ### Author Response · Authors · 2022-11-18
> > **Re: Questions about experiments on WikiText-103**
> >
> > Thank you for your question. There are a few differences between the Transformer-XL models and standard GPT-style Transformers that make direct comparison difficult:
> > * The Transformer-XL models use a different tokenizer than the GPT-2/GPT-3/Neo models. The Transformer-XL tokenizer has a different vocab size, so the PPLs are not directly comparable.
> > * The Transformer-XL vocab size requires a change to the model (adaptive softmax), which may affect performance.
> >
> > In our models, we follow many recent large language models (GPT2, GPT-3, GPT-Neo, Gopher, Chinchilla, etc) in using the GPT-2 tokenizer for standardization. The top five numbers on the table are all trained with the same tokenizer, with the only change being the attention/H3 module – so those numbers are directly comparable to each other.
> >
> > We have added a discussion of these details to the Appendix.

---

### Author Response · Authors · 2022-12-08
**Update and Gentle Reminder**

We would like to thank everyone for their constructive feedback again. We would like to give a gentle reminder that the discussion period is ending soon. We hope that we have addressed all reviewers' concerns, and we would be happy to discuss any further concerns!

We would also like to provide a new update. With FlashFFTConv, we have been able to scale our model to 1.3 billion parameters, trained on the Pile for 300B tokens with positive results.

On val PPL, our 1.3B H3 model outperforms GPT-Neo 1.3B (which is also trained on the Pile):

| **Model**  | **Pile PPL** |
| ---: | :---: |
| GPT-Neo 1.3B | 7.4 |
| Hybrid H3 1.3B | 6.2 |

On downstream SuperGLUE evaluation, our 1.3B model outperforms GPT-Neo 1.3B and OPT 1.3B on nearly all tasks in zero-shot learning and outperforms the other models on average in few-shot learning.

**Zero Shot:**
| **Model**  | **WSC** | **WIC** | **RTE** | **CB** | **MultiRC** | **ReCoRD** | **BoolQ** | **COPA** | **Avg** |
| ---: | :---: | :---: | :---: | :---: | :---: | :---: | :---: | :---: | :---: |
| OPT 1.3B | 36.5 | 49.5 | **53.4** | **39.3** | **58.3** | 61.8 | 55.0 | 69.0 | 52.9 |
| GPT-Neo 1.3B | 41.3 | 50.0 | 52.3 | 33.9 | 57.9 | 55.5 | 59.9 | 66.0 | 52.1 |
| Hybrid H3 1.3B | **52.9** | **50.3** | **53.4** | 33.9 | 58.2 | **67.8** | **61.7** | **74** | **56.5** |

**Few Shot:**
| **Model**  | **WSC** | **WIC** | **RTE** | **CB** | **MultiRC** | **ReCoRD** | **BoolQ** | **COPA** | **Avg** |
| ---: | :---: | :---: | :---: | :---: | :---: | :---: | :---: | :---: | :---: |
| OPT 1.3B | **44.2** | **51.1** | 53.4 | 16.1 | **59.9** | 62.1 | 38.3 | 70.0 | 49.4 |
| GPT-Neo 1.3B | 35.6 | 50.6 | 47.3 | **32.1** | **59.9** | 55.7 | **61.2** | 67.0 | 51.2 |
| Hybrid H3 1.3B | 36.5 | 49.2 | **55.2** | 23.2 | 59.3 | **67.6** | 56.9 | **76.0** | **53.0** |

These results validate that H3 may scale better than Transformers in billion+ parameter model sizes, and we’re excited to see how it continues to scale in the future.

We are happy to discuss any further concerns that any reviewers may have!

---

### Decision · Program_Chairs · 2023-01-20

**Decision:**

Accept: notable-top-25%

**Justification For Why Not Higher Score:**

Stronger overall results.

**Justification For Why Not Lower Score:**

Useful paper for the community, and the authors have managed to verify it on a 1.3B model.

**Metareview: Summary, Strengths And Weaknesses:**

This paper presents approaches to improve state space models for language models. I think this is a very interesting study and it is great that the proposed method is designed based on preliminary experiments on synthetic data to understand the limitations of SSMs. The results are promising and a step in the right direction for this class of models. All reviewers agree that this is a good paper. I recommend acceptance.

**Note From Pc:**

if the above contains the word "oral" or "spotlight" please see: "oral" presentation means -> notable-top-5% and "spotlight" means -> notable-top-25%. As stated in our emails, we are disassociating presentation type from AC recommendations